# On Theoretical Limits of Learning with Label Differential Privacy

## Abstract

Label differential privacy (DP) is designed for learning problems with private labels and public features. Although various methods have been proposed for learning under label DP, the theoretical limits remain unknown. The main challenge is to take infimum over all possible learners with arbitrary model complexity. In this paper, we investigate the fundamental limits of learning with label DP under both central and local models. To overcome the challenge above, we derive new lower bounds on testing errors that are adaptive to the model complexity. Our analyses indicate that $\epsilon$-local label DP only enlarges the sample complexity with respect to $\epsilon$, without affecting the convergence rate over the sample size $N$, except the case with heavy-tailed label. Under the central model, the performance loss due to the privacy mechanism is further weakened, such that the additional sample complexity becomes negligible. Overall, our analysis validates the promise of learning under the label DP from a theoretical perspective and shows that the learning performance can be significantly improved by weakening the DP definition to only labels.

## 1 Introduction

Many modern machine learning tasks require sensitive training samples that need to be protected from leakage [1]. As a standard approach for privacy protection, differential privacy (DP) [2] has been extensively studied [3–9]. However, the learning performances under original DP definition are usually far from satisfactory [10–13]. Therefore, researchers attempt to design weakened DP requirements, under which the performances can be significantly improved, while still securing sensitive information. Under such background, label DP has emerged in recent years [14], which regards features as public, while only labels are sensitive and need to be protected. Such setting is realistic in many applications, such as computational advertising [15], recommendation systems [16] and medical diagnosis [17]. These tasks usually use some basic demographic information as features, which can be far less sensitive.

Despite various approaches for learning with label DP [14, 18–21], the fundamental limits are still unknown. An interesting question is: By weakening the DP definitions to only labels, how much accuracy improvement is possible? From an information-theoretic perspective [22], the underlying limits of statistical problems are characterized by the minimax lower bound, which takes the supremum over all possible distributions from a general class, and infimum over all learners. Deriving minimax lower bounds for learning under the label DP is challenging in two aspects. Firstly, under label DP, each sample has both public (i.e. the feature) and private (i.e. the label) components. Directly applying the methods for original DP [23–27] treats all components as private, and thus does not yield tight results. Secondly, the classical packing method [47] is only suitable for fixed model structures with fixed dimensionality. However, to establish lower bounds, one needs to take infimum over all possible learners with arbitrary model complexity.

Submitted to 38th Conference on Neural Information Processing Systems (NeurIPS 2024). Do not distribute.

| | Classification | Regression Bounded label noise | Regression Unbounded label noise |
|---|---|---|---|
| Local | $\tilde{O}((N(\epsilon^2 \wedge 1))^{-\frac{\beta(\gamma+1)}{2\beta+d}})$ | $\tilde{O}\left((N(\epsilon^2 \wedge 1)^{-\frac{2\beta}{d+2\beta}})\right)$ | $O\left((N\epsilon^2)^{-\frac{2\beta(p-1)}{2p\beta+d(p-1)}} \vee N^{-\frac{2\beta}{2\beta+d}}\right)$ |
| Central | $\tilde{O}\left(N^{-\frac{\beta(\gamma+1)}{2\beta+d}} + (\epsilon N)^{-\frac{\beta(\gamma+1)}{\beta+d}}\right)$ | $O\left(N^{-\frac{2\beta}{2\beta+d}} + (\epsilon N)^{-\frac{2\beta}{d+\beta}}\right)$ | $O\left(N^{-\frac{2\beta}{2\beta+d}} + (\epsilon N)^{-\frac{2\beta(p-1)}{p\beta+d(p-1)}}\right)$ |
| Local full | $O((N(\epsilon^2 \wedge 1))^{-\frac{\beta(\gamma+1)}{2\beta+2d}})$ | $O((N(\epsilon^2 \wedge 1))^{-\frac{\beta}{\beta+d}})$ | $O((N(\epsilon^2 \wedge 1))^{-\frac{\beta(p-1)}{p\beta+d(p-1)}})$ |
| Non-priv. | $O(N^{-\frac{\beta(\gamma+1)}{2\beta+d}})$ | $O(N^{-\frac{2\beta}{2\beta+d}})$ | $O(N^{-\frac{2\beta}{2\beta+d}})$ |

Table 1: Minimax rate of convergence under label differential privacy. $d$ is the dimension of features.

In this paper, we investigate the theoretical limits of classification and regression problems under label DP. Our analysis involves both central and local models. For each problem, we derive the information-theoretic minimax lower bound of the risk function over a wide class of distributions satisfying the $\beta$-Hölder smoothness and the $\gamma$-Tsybakov margin assumption [28] (see Assumption 1 for details). The general idea is to convert the problem to multiple hypothesis testing. To overcome the challenges above, we provide a bound of Kullback-Leibler divergence over joint distributions of private and public random variables, which is tighter than the bound between fully private variables. Moreover, under the central model, instead of using the packing method, we develop a new lower bound on the minimum testing error for each pair of hypotheses based on the group privacy property [4], which is suitable for arbitrary model complexity. After deriving minimax lower bounds, we also propose algorithms with matching upper bounds to validate the tightness of our results.

The results are shown in Table 1, in which the third row refers to the bounds under the original local DP definition, while the fourth row lists the non-private baselines. To the best of our knowledge, minimax rates under central DP have not been established, and are thus not listed here. The main findings are summarized as follows.

- Under $\epsilon$-local label DP, for classification and regression with bounded label noise, the sample complexity is larger by a factor of $O(1/\epsilon^2)$. However, the convergence rate remains unaffected, which is in clear contrast with the original DP, under which the convergence rate is slower.

- Under $\epsilon$-local label DP constraint, for regression with heavy-tailed label noise, the convergence rate of risk over $N$ becomes slower, indicating that heavy-tailed labels increase the difficulty of privacy protection.

- Under $\epsilon$-central label DP constraint, the performance loss caused by the privacy mechanism becomes further weakened. The risk only increases by a term that decays faster than the non-private rate, indicating that the additional sample complexity caused by the privacy mechanism becomes negligible with large $N$.

In general, our analysis provides a theoretical perspective of understanding label DP. The result shows that by weakening the DP definition to protecting labels only, the learning performances can be significantly improved.

## 2  Related Work

**Label DP.** Under the local model, labels are randomized before training. The simplest method is randomized response [30]. An important improvement is proposed in [14], called RRWithPrior, which incorporates prior distribution. [19] proposes ALIBI, which further improves randomized response by generating soft labels through Bayesian inference. There are also several methods for regression under label DP [18, 31]. Under central label DP, [20] proposes a clustering approach. [19] proposes private aggregation of teacher ensembles (PATE), which is then further improved in [21].

**Minimax analysis for public data.** Minimax theory provides a rigorous framework for the best possible performance of an algorithm given some assumptions. Classical methods include Le Cam [32], Fano [33] and Assouad [34]. Using these methods, minimax lower bounds have been widely established for both classification and regression problems [28, 29, 35–41]. If the feature vector has bounded support, then the minimax rate of classification and regression are $O(N^{-\frac{\beta(\gamma+1)}{2\beta+d}})$ and $O(N^{-\frac{2\beta}{2\beta+d}})$, respectively.

**Minimax analysis for private data.** Under the local model, [42] finds the relation between label DP and stochastic query. [23] and [24] develop the variants of Le Cam, Fano, and Assouad's method under local DP. Lower bounds are then established for various statistical problems, such as mean estimation [43–46], classification [26] and regression [27]. Under central model, for pure DP, the standard approach is the packing method [47], which is then used in hypothesis testing [48], mean estimation [49,50], and learning of distributions [51–53]. There are also several works on approximate DP, such as [54, 55].

This work studies the theoretical limits of label DP, under which each sample is a mixture of public feature and private labels, thus existing methods can not be directly applied here. Under the central model, the minimax analysis becomes more challenging, since the packing method is only suitable for fixed model structures (i.e. the dimensionality of model output is fixed), while we need to find the minimum possible error over all possible learners with arbitrary output dimensions. As a result, the lower bounds of general classification and regression problems have not been established even under the original DP definition. To overcome such challenge, we develop a new approach to bound the error of hypothesis testing (see Lemma 1 in Appendix D).

## 3 Preliminaries

In this section, we show some necessary definitions, background information, and notations.

### 3.1 Label DP

To begin with, we review the definition of DP. Suppose the dataset consists of $N$ samples $(\mathbf{x}_i, y_i)$, $i = 1, \ldots, N$, in which $\mathbf{x}_i \in \mathcal{X}$ is the feature vector, while $y_i \in \mathcal{Y} \subset \mathbb{R}^d$ is the label.

**Definition 1.** *(Differential Privacy (DP) [2]) Let $\epsilon \geq 0$. A randomized function $\mathcal{A} : (\mathcal{X}, \mathcal{Y})^N \to \Theta$ is $\epsilon$-DP if for any two adjacent datasets $D, D' \in (\mathcal{X}, \mathcal{Y})^N$ and any $S \subseteq \Theta$,*

$$P(\mathcal{A}(D) \in S) \leq e^\epsilon P(\mathcal{A}(D') \in S), \tag{1}$$

*in which $D$ and $D'$ are adjacent if they differ only on a single sample, including both the feature vector and the label.*

In machine learning tasks, the output of $\mathcal{A}$ is the model parameters, while the input is the training dataset. Definition 1 requires that both features and labels are privatized. Consider that in some applications, the features may be much less sensitive, the notion of label DP is defined as follows.

**Definition 2.** *(Central label DP) A randomized function $\mathcal{A}$ is $\epsilon$-label DP if for any two datasets $D$ and $D'$ that differ on the label of only one training sample and any $S \subseteq \Theta$, (1) holds.*

Compared with Definition 1, Definition 2 only requires the output to be insensitive to the replacement of a label. Therefore label DP is a weaker requirement. Correspondingly, the local label DP is defined as follows.

**Definition 3.** *(Local label DP) A randomized function $M : (\mathcal{X}, \mathcal{Y}) \to \mathcal{Z}$ is $\epsilon$-local label DP if*

$$\sup_{y, y' \in \mathcal{Y}} \sup_{S \subseteq \mathcal{Z}} \ln \frac{P(M(\mathbf{x}, y) \in S)}{P(M(\mathbf{x}, y') \in S)} \leq \epsilon. \tag{2}$$

Definition 3 requires that each label is privatized locally before running any machine learning algorithms. It is straightforward to show that local label DP ensures central label DP. To be more precise, we have the following proposition.

**Proposition 1.** *Let $\mathbf{z}_i = M(\mathbf{x}_i, y_i)$ for $i = 1, \ldots, N$. If $\mathcal{A}$ is a function of $(\mathbf{x}_i, \mathbf{z}_i)$, $i = 1, \ldots, N$, then $\mathcal{A}$ is $\epsilon$-label DP.*

### 3.2 Risk of Classification and Regression

In supervised learning problems, given $N$ samples $(\mathbf{X}_i, Y_i)$, $i = 1, \ldots, N$ drawn from a common distribution, the task is to learn a function $g : \mathcal{X} \to \mathcal{Y}$. For a loss function $l : \mathcal{Y} \times \mathcal{Y} \to \mathbb{R}$, the goal is to minimize the *risk function*, which is defined as the expectation of loss function between the predicted value and the ground truth:

$$R = \mathbb{E}[l(\hat{Y}, Y)]. \tag{3}$$

The minimum risk among all function $g$ is called Bayes risk, i.e. $R^* = \min_g \mathbb{E}[l(g(\mathbf{X}, Y))]$. In practice, the sample distribution is unknown, and we need to learn $g$ from samples. Therefore, the risk of any practical classifiers is larger than Bayes risk. The gap $R - R^*$ is called excess risk, and we hope that $R - R^*$ to be as small as possible. Now we discuss classification and regression problems separately.

*1) Classification.* For classification problems, the size of $\mathcal{Y}$ is finite. For convenience, we denote $\mathcal{Y} = [K]$, in which $[K] := \{1, \ldots, K\}$. In this paper, we use $0 - 1$ loss, i.e. $l(\hat{Y}, Y) = \mathbf{1}(\hat{Y} \neq Y)$, then $R = \mathrm{P}(\hat{Y} \neq Y)$. Define $K$ functions $\eta_1, \ldots, \eta_K$ as the conditional class probabilities:

$$\eta_k(\mathbf{x}) = \mathrm{P}(Y = k | \mathbf{X} = \mathbf{x}), k = 1, \ldots, K. \tag{4}$$

Under this setting, the Bayes optimal classifier and the corresponding Bayes risk is

$$c^*(\mathbf{x}) = \arg\max_{j \in [K]} \eta_j(\mathbf{x}), \tag{5}$$

$$R^*_{cls} = \mathrm{P}(c^*(\mathbf{X}) \neq Y). \tag{6}$$

*2) Regression.* Now we consider the case with $\mathcal{Y}$ having infinite size. We use $\ell_2$ loss in this paper, i.e. $l(\hat{Y}, Y) = (\hat{Y} - Y)^2$. Then the Bayes risk is

$$R^*_{reg} = \mathbb{E}[(Y - \eta(\mathbf{X}))^2]. \tag{7}$$

Then the following proposition gives a bound of the excess risk for classification and regression problems.

**Proposition 2.** *For any classifier $c : \mathcal{X} \to [K]$, the excess risk of classification is bounded by*

$$R_{cls} - R^*_{cls} = \int (\eta^*(\mathbf{x}) - \mathbb{E}[\eta_{c(\mathbf{x})}(\mathbf{x})]) f(\mathbf{x}) d\mathbf{x}. \tag{8}$$

*For any regression estimate $\hat{\eta} : \mathcal{X} \to \mathcal{Y}$, the excess risk of regression is bounded by*

$$R_{reg} - R^*_{reg} = \mathbb{E}[(\hat{\eta}(\mathbf{X}) - \eta(\mathbf{X}))^2]. \tag{9}$$

The proof of Proposition 2 is shown in Appendix A. Finally, we state some basic assumptions that will be used throughout this paper.

**Assumption 1.** *There exists some constants $L$, $\beta$, $C_T$, $\gamma$, $c$, $D$ and $\theta \in (0, 1]$ such that*

*(a) For all $j \in [K]$ and any $\mathbf{x}, \mathbf{x}'$, $|\eta_j(\mathbf{x}) - \eta_j(\mathbf{x}')| \leq L \|\mathbf{x} - \mathbf{x}'\|^\beta$;*

*(b) For any $t > 0$, $P(0 < \eta^*(\mathbf{X}) - \eta_s(\mathbf{X}) < t) \leq C_T t^\gamma$, in which $\eta_s(\mathbf{x})$ is the second largest one among $\{\eta_1(\mathbf{x}), \ldots, \eta_K(\mathbf{x})\}$;*

*(c) The feature vector $\mathbf{X}$ has a probability density function (pdf) $f$ which is bounded from below, i.e. $f(\mathbf{x}) \geq c$;*

*(d) For all $r < D$, $V_r(\mathbf{x}) \geq \theta v_d r^d$, in which $V_r(\mathbf{x})$ is the volume (Lebesgue measure) of $B(\mathbf{x}, r) \cap \mathcal{X}$, $v_d$ is the volume of a unit ball.*

Assumption 1 (a) requires that all $\eta_j$ are Hölder continuous. This condition is common in literatures about nonparametric statistics [28]. (b) is generalized from the Tsybakov noise assumption for binary classification, which is commonly used in many existing works in the field of both nonparametric classification [29, 37, 40, 41] and differential privacy [26, 27]. If $K = 2$, then $\eta^*$ and $\eta_s$ refer to the larger and smaller class conditional probability, respectively. An intuitive understanding of (b) is that in the majority of the support, the maximum value among $\{\eta_1(\mathbf{x}), \ldots, \eta_K(\mathbf{x})\}$ should have some gap to the second largest one. With sufficiently large sample size and model complexity, assumption (b) ensures that for test samples within the majority of the support $\mathcal{X}$, the algorithm is highly likely to correctly identify the class with the maximum conditional probability. Therefore, in (b), we only care about $\eta^*(\mathbf{x})$ and $\eta_s(\mathbf{x})$, while other classes with small conditional probabilities can be ignored. (c) is usually called "strong density assumption" in existing works [39, 40], which is quite strong. It is possible to relax this assumption so that the theoretical analysis becomes suitable for general cases. However, we do not focus on such generalization in this paper. Assumption (d) prevents the corner of the support $\mathcal{X}$ from being too sharp. In the remainder of this section, denote $\mathcal{F}_{cls}$ as the set of all pairs $(f, \eta)$ satisfying Assumption 1.

 # 4    Classification

 In this section, we derive the upper and lower bounds of learning under central and local label DP,
 respectively.

## 4.1    Local Label DP

 *1) Lower bound.* The following theorem shows the minimax lower bound, which characterizes the
 theoretical limit.

**Theorem 1.** *Denote $\mathcal{M}_\epsilon$ as the set of all privacy mechanisms satisfying $\epsilon$-local label DP (Definition 3). Then*

$$\inf_{\hat{Y}} \inf_{M \in \mathcal{M}_\epsilon} \sup_{(f,\eta) \in \mathcal{F}_{cls}} (R_{cls} - R_{cls}^*) \gtrsim \left[ N \left( \epsilon^2 \wedge 1 \right) \right]^{-\frac{\beta(\gamma+1)}{2\beta+d}}. \tag{10}$$

 *Proof.* (Outline) It suffices to derive (10) with $K = 2$. We convert the problem into multiple binary
 hypothesis testing problems. In particular, we divide the support into $G$ bins. For some of them, we
 construct two opposite hypotheses such that they are statistically not distinguishable. Our proof uses
 some techniques in local DP [24] and some classical minimax theory [28]. The detailed proof is
 shown in Appendix B.    □

 In Theorem 1, (10) takes supremum over all joint distributions of $(\mathbf{X}, Y)$, and infimum over all
 classifiers and privacy mechanisms satisfying $\epsilon$-local label DP.

 *2) Upper bound.* We then show that the bound (10) is achievable. Let the privacy mechanism $M(\mathbf{x}, y)$
 outputs a $K$ dimensional vector, with each component being either 0 or 1, such that

$$\mathrm{P}(M(\mathbf{x}, y)(j) = 1) = \begin{cases} \frac{e^{\frac{\epsilon}{2}}}{e^{\frac{\epsilon}{2}}+1} & \text{if} \quad y = j \\ \frac{1}{e^{\frac{\epsilon}{2}}+1} & \text{if} \quad y \neq j, \end{cases} \tag{11}$$

 and $\mathrm{P}(M(\mathbf{x}, y)(j) = 0) = 1 - \mathrm{P}(M(\mathbf{x}, y)(j) = 1)$, in which $M(\mathbf{x}, y)(j)$ is the $j$-th component of
 $M(\mathbf{x}, y)$. For $N$ random training samples $(\mathbf{X}_i, Y_i)$, let $\mathbf{Z}_i = M(\mathbf{X}_i, Y_i)$, and correspondingly, $Z_i(j)$
 is the $j$-th component of $\mathbf{Z}_i$.

 Divide the support $\mathcal{X}$ into $G$ bins, named $B_1, \ldots, B_G$, such that the length of each bin is $h$.
 $B_1, \ldots, B_G$ are disjoint, and these bins form a covering of $\mathcal{X}$, i.e. $\mathcal{X} \subset \cup_{l=1}^G B_l$. Then calcu-
 late

$$S_{lj} = \sum_{i:\mathbf{X}_i \in B_l} Z_i(j), l = 1, \ldots, G, j = 1, \ldots, K, \tag{12}$$

 The classification within the $l$-th bin is

$$c_l = \arg\max_j S_{lj}, \tag{13}$$

 such that the the prediction given $\mathbf{x}$ is $c(\mathbf{x}) = c_l$ for all $\mathbf{x} \in B_l$. The next theorem shows the privacy
 guarantee, as well as the bound of the excess risk.

**Theorem 2.** *The privacy mechanism $M$ is $\epsilon$-local label DP. Moreover, under Assumption 1, with* $h \sim \left( N(\epsilon^2 \wedge 1)/\ln K \right)^{-\frac{1}{2\beta+d}}$, *the excess risk of the classifier described above can be upper bounded as follows:*

$$R_{cls} - R_{cls}^* \lesssim \left( \frac{N(\epsilon^2 \wedge 1)}{\ln K} \right)^{-\frac{\beta(\gamma+1)}{2\beta+d}}. \tag{14}$$

 *Proof.* (Outline) For privacy guarantee, we need to show that (11) is $\epsilon$-local label DP:

$$\begin{aligned} \frac{\mathrm{P}(M(\mathbf{x}, y) = \mathbf{z})}{\mathrm{P}(M(\mathbf{x}, y') = \mathbf{z})} &= \Pi_{j=1}^K \frac{\mathrm{P}(M(\mathbf{x}, y)(j) = \mathbf{z}(j))}{\mathrm{P}(M(\mathbf{x}, y')(j) = \mathbf{z}(j))} \\ &= \frac{\mathrm{P}(M(\mathbf{x}, y)(y) = \mathbf{z}(y))}{\mathrm{P}(M(\mathbf{x}, y')(y) = \mathbf{z}(y))} \frac{\mathrm{P}(M(\mathbf{x}, y)(y') = \mathbf{z}(y'))}{\mathrm{P}(M(\mathbf{x}, y')(y') = \mathbf{z}(y'))} \\ &\leq e^{\frac{\epsilon}{2}} e^{\frac{\epsilon}{2}} = e^\epsilon. \end{aligned} \tag{15}$$

According to Definition 3, $M$ is $\epsilon$-local label DP. For the performance guarantee (14), according to Proposition 2, we need to bound $\eta^*(\mathbf{x}) - \mathbb{E}[\eta_{c(\mathbf{x})}(\mathbf{x})]$ for each $\mathbf{x}$. If $\eta^*(\mathbf{x}) - \eta_s(\mathbf{x})$ is large, then with high probability, $c(\mathbf{x}) = c^*(\mathbf{x})$, and then $\eta^*(\mathbf{x}) = \eta_{c(\mathbf{x})}(\mathbf{x})$. Thus we mainly consider the case with small $\eta^*(\mathbf{x}) - \eta_s(\mathbf{x})$. The details of proof are shown in Appendix C. $\qquad\square$

The lower bound (10) and the upper bound (14) match up to a logarithm factor, indicating that the results are tight. Now we comment on the results.

**Remark 1.** *1) **Comparison with non-private bound.** The classical minimax lower bound for non-private classification problem is $N^{-\frac{\beta(\gamma+1)}{2\beta+d}}$. Therefore, the lower bound (10) reaches the non-private bound with $\epsilon \gtrsim 1$. With small $\epsilon$, $N$ training samples with privatized labels roughly equals $N\epsilon^2$ non-privatized samples in terms of performance.*

*2) **Comparison with local DP that protects both features and labels.** In this case, the optimal excess risk is $(N\epsilon^2)^{-\beta(\gamma+1)/(2\beta+2d)} \vee N^{-\beta(\gamma+1)/(2\beta+d)}$, which is worse than the right hand side of (10). Such result indicates that compared with classical DP, label DP incurs significantly weaker performance loss.*

*3) **Comparison with other baseline methods.** If we use the randomized response method instead of the privacy mechanism (11), then the performance decreases sharply with the number of classes $K$. Several methods have been proposed to improve the randomized response method, such as RRWithPrior [14] and ALIBI [19]. However, these methods are not guaranteed in theory.*

## 4.2 Central Label DP

*1) Lower bound.* The following theorem shows the minimax lower bound under the central label DP.

**Theorem 3.** *Denote $\mathcal{A}_\epsilon$ as the set of all learning algorithms satisfying $\epsilon$-label DP (Definition 2). Then*

$$\inf_{\mathcal{A} \in \mathcal{A}_\epsilon} \sup_{(f,\eta) \in \mathcal{F}_{cls}} (R_{cls} - R_{cls}^*) \gtrsim N^{-\frac{\beta(\gamma+1)}{2\beta+d}} + (\epsilon N)^{-\frac{\beta(\gamma+1)}{\beta+d}}. \tag{16}$$

*Proof.* (Outline) Lower bounds under central DP are usually constructed by packing method [47], which works for fixed output dimensions. However, to achieve a desirable bias and variance tradeoff, the model complexity needs to increase with $N$. In our proof, we still divide the support into $G$ bins and construct two hypotheses for each bin, but we develop a new tool (see Lemma 1) to give a lower bound of the minimum error of hypothesis testing. We then use the group privacy property [4] to get the overall lower bound. The details can be found in Appendix D. $\qquad\square$

*2) Upper bound.* Now we show that (16) is achievable. Similar to the local label DP problem, now divide the support into $G$ bins, such that the length of each bin is $h$. Now the classification within the $l$-th bin follows a exponential mechanism [56]:

$$\mathrm{P}(c_l = j | \mathbf{X}_{1:N}, Y_{1:N}) = \frac{e^{\epsilon n_{lj}/2}}{\sum_{k=1}^K e^{\epsilon n_{lk}/2}}, \tag{17}$$

in which $n_{lj} = \sum_{i=1}^N \mathbf{1}(\mathbf{X}_i \in B_l, Y_i = j)$. Then let $c(\mathbf{x}) = c_l$ for $\mathbf{x} \in B_l$. The excess risk is bounded in the next theorem.

**Theorem 4.** *The privacy mechanism (17) is $\epsilon$-label DP. Moreover, under Assumption 1, if $h$ scales as $h \sim (\ln K/\epsilon N)^{\frac{1}{\beta+d}} + (\ln K/N)^{\frac{1}{2\beta+d}}$, then the excess risk can be bounded as follows:*

$$R - R^* \lesssim \left(\frac{N}{\ln K}\right)^{-\frac{\beta(\gamma+1)}{2\beta+d}} + \left(\frac{\epsilon N}{\ln K}\right)^{-\frac{\beta(\gamma+1)}{\beta+d}}. \tag{18}$$

*Proof.* (Outline) The privacy guarantee of the exponential mechanism has been analyzed in [4]. Following these existing analyses, it can be shown that (17) is $\epsilon$-label DP. It remains to show (18). Note that if $\eta^*(\mathbf{x}) - \eta_s(\mathbf{x})$ is large, then the difference between the largest and the second largest one from $\{n_{lj}|j = 1, \ldots, K\}$ will also be large. From (17), the following inequality holds with high probability: $c_l = \arg\max_j n_{lj} = \arg\max_j \eta_j(\mathbf{x}) = c^*(\mathbf{x})$, which means that the classifier makes

optimal prediction. Hence we mainly consider the case with small $\eta^*(\mathbf{x}) - \eta_s(\mathbf{x})$. The details of the proof can be found in Appendix E. □

The upper and lower bounds match up to logarithmic factors. In (18), the first term is just the non-private convergence rate, while the second term $(\epsilon N)^{-\frac{\beta(\gamma+1)}{\beta+d}}$ can be regarded as the additional risk caused by the privacy mechanism. It decays faster with $N$ compared with the first term, thus the additional performance loss caused by the privacy mechanism becomes negligible as $N$ increases. This result is crucially different from the local model, under which the privacy mechanism always induces higher sample complexity by a factor of $O(1/(\epsilon^2 \wedge 1))$.

## 5   Regression with Bounded Noise

Now we analyze the theoretical limits of regression problems under local and central label DP. Throughout this section, we assume that the label is restricted within a bounded interval.

**Assumption 2.** *Given any $\mathbf{x} \in \mathcal{X}$, $P(|Y| < T | \mathbf{X} = \mathbf{x}) = 1$.*

Assumption 1 remains the same here. In the remainder of this section, denote $\mathcal{F}_{reg1}$ as the set of $(f, \eta)$ that satisfies Assumption 1 and 2.

### 5.1   Local Label DP

*1) Lower bound.* Theorem 5 shows the minimax lower bound.

**Theorem 5.** *Denote $\mathcal{M}_\epsilon$ as the set of all privacy mechanisms satisfying $\epsilon$-label DP. Then*

$$\inf_{\hat{\eta}} \inf_{M \in \mathcal{M}_\epsilon} \sup_{(f,\eta) \in \mathcal{F}_{reg1}} (R_{reg} - R_{reg}^*) \gtrsim (N(\epsilon^2 \wedge 1))^{-\frac{2\beta}{d+2\beta}}. \tag{19}$$

The proof of Theorem 5 is similar to that of Theorem 1, except for some details in hypotheses construction and the final bound of excess risk. The details are shown in Appendix F.

*2) Upper bound.* The privacy mechanism is $Z = Y + W$, in which $W \sim \mathrm{Lap}(2T/\epsilon)$. Then the privacy mechanism satisfies $\epsilon$-label DP. In this case, the real regression function $\eta(\mathbf{x})$ can be estimated using the nearest neighbor approach. Let

$$\hat{\eta}(\mathbf{x}) = \frac{1}{k} \sum_{i \in \mathcal{N}_k(\mathbf{x})} Z_i, \tag{20}$$

in which $\mathcal{N}_k(\mathbf{x})$ is the set of $k$ nearest neighbors of $\mathbf{x}$ among $\mathbf{X}_1, \ldots, \mathbf{X}_N$.

**Theorem 6.** *The method described above is $\epsilon$-local label DP. Moreover, with $k \sim N^{\frac{2\beta}{d+2\beta}}(\epsilon \wedge 1)^{-\frac{2d}{d+2\beta}}$, then under Assumption 1 and 2,*

$$R_{reg} - R_{reg}^* \lesssim (N(\epsilon^2 \wedge 1))^{-\frac{2\beta}{d+2\beta}}. \tag{21}$$

*Proof.* (Outline) Since $|Y| < T$, $W \sim \mathrm{Lap}(2T/\epsilon)$, it is obvious that $Z = Y + W$ is $\epsilon$-local label DP. For the performance (21), the bias can be bounded by the $k$ nearest neighbor distances based on Assumption 1(a). The variance of $\hat{\eta}(\mathbf{x})$ scales inversely with $k$. An appropriate $k$ can be selected to achieve a good tradeoff between bias and variance. The details are shown in Appendix G. □

From standard minimax analysis on regression problems, the non-private convergence rate is $N^{-2\beta/(d+2\beta)}$. From Theorem 5 and 6, the privatization process makes sample complexity larger by a $O(1/\epsilon^2)$ factor.

### 5.2   Central Label DP

*1) Lower bound.* The following theorem shows the minimax lower bound.

**Theorem 7.** *Let $\mathcal{A}_\epsilon$ be the set of all algorithms satisfying $\epsilon$-central DP. Then*

$$\inf_{\mathcal{A} \in \mathcal{A}_\epsilon} \sup_{(f,\eta) \in \mathcal{F}_{reg1}} (R_{reg} - R_{reg}^*) \gtrsim N^{-\frac{2\beta}{2\beta+d}} + (\epsilon N)^{-\frac{2\beta}{d+\beta}}. \tag{22}$$

267   *2) Upper bound.* For each bin $B_l$, let $n_l = \sum_{i=1}^{N} \mathbf{1}(\mathbf{X}_i \in B_l)$ be the number of samples in $B_l$. If
268   $n_l > 0$, then

$$\hat{\eta}_l = \frac{1}{n_l} \sum_{i=1}^{N} \mathbf{1}(\mathbf{X}_i \in B_l) Y_i + W_l, \tag{23}$$

269   in which $W_l \sim \text{Lap}(2/(n_l \epsilon))$. If $n_l = 0$, i.e. no sample falls in $B_l$, then just let $\hat{\eta}_l = 0$. For all
270   $\mathbf{x} \in B_l$, let $\hat{\eta}(\mathbf{x}) = \hat{\eta}_l$. The excess risk can be bounded with the following theorem.

271   **Theorem 8.** *(23) is $\epsilon$-label DP. Moreover, under Assumption 1 and 2, if $h$ scales as $h \sim N^{-\frac{1}{2\beta+d}} +$*
272   $(\epsilon N)^{-\frac{1}{d+\beta}}$, *then the excess risk is bounded by*

$$R - R^* \lesssim N^{-\frac{2\beta}{2\beta+d}} + (\epsilon N)^{-\frac{2\beta}{d+\beta}}. \tag{24}$$

273   The upper and lower bounds match, indicating that the results are tight. Again, the second term in
274   (24) converges faster than the first one with respect to $N$, the performance loss caused by privacy
275   constraints becomes negligible as $N$ increases.

## 6   Regression with Heavy-tailed Noise

277   In this section, we consider the case such that the noise has tails. We make the following assumption.

278   **Assumption 3.** *For all $\mathbf{x} \in \mathcal{X}$, $\mathbb{E}[|Y|^p | \mathbf{X} = \mathbf{x}] \leq M_p$ for some $p \geq 2$.*

279   Instead of requiring $|Y| < T$ for some $T$, now we only assume that the $p$-th order moment is bounded.
280   For non-private cases, given fixed noise variance, the tail does not affect the mean squared error of
281   regression. As a result, as long as $p \geq 2$, the convergence rate of regression risk is the same as the
282   case with bounded noise. However, the label DP requires the output to be insensitive to the worst
283   case replacement of labels, which can be harder if the noise has tails. To achieve $\epsilon$-DP, the clipping
284   radius decreases with $\epsilon$, thus the noise strength needs to grow faster than $O(1/\epsilon)$. As a result, the
285   convergence rate becomes slower than the non-private case. In the remainder of this section, denote
286   $\mathcal{F}_{reg2}$ as the set of $(f, \eta)$ that satisfies Assumption 1 and 3.

### 6.1   Local Label DP

288   *1) Lower bound.* In earlier sections about classification and regression with bounded noise, the impact
289   of privacy mechanisms is only a polynomial factor on $\epsilon$, while the convergence rate of excess risk
290   with respect to $N$ is not changed. However, this rule no longer holds when the noise has heavy tails.

291   **Theorem 9.** *Denote $\mathcal{M}_\epsilon$ as the set of all privacy mechanisms satisfying $\epsilon$-label DP. Then for small $\epsilon$,*

$$\inf_{\hat{\eta}} \inf_{M \in \mathcal{M}_\epsilon} \sup_{(f,\eta) \in \mathcal{F}} (R_{reg} - R_{reg}^*) \gtrsim (N(e^\epsilon - 1)^2)^{-\frac{2\beta(p-1)}{2p\beta+d(p-1)}} + N^{-\frac{2\beta}{2\beta+d}}. \tag{25}$$

292   *2) Upper bound.* Since now the noise has unbounded distribution, without preprocessing, the
293   sensitivity is unbounded, thus simply adding noise to $Y$ can no longer protect the privacy. Therefore,
294   a solution is to clip $Y$ into $[-T, T]$, and add noise proportional to $T/\epsilon$ to achieve $\epsilon$-local label DP.
295   Such truncation will inevitably introduce some bias. To achieve a tradeoff between clipping bias and
296   sensitivity, the value of $T$ needs to be tuned carefully. Based on such intuition, the method is precisely
297   stated as follows. Let $Z_i = Y_{Ti} + W_i$, in which $Y_{Ti}$ is the truncation of $Y_i$, i.e. $Y_{Ti} = (Y_i \wedge T) \vee (-T)$,
298   and $W \sim \text{Lap}(2T/\epsilon)$. The result is shown in the next theorem.

299   **Theorem 10.** *The method above is $\epsilon$-local label DP. Moreover, with $k \sim (N\epsilon^2)^{\frac{2p\beta}{2p\beta+d(p-1)}} \vee N^{\frac{2\beta}{2\beta+d}}$,*
300   *and $T \sim (k\epsilon^2)^{\frac{1}{2p}}$, the risk is bounded by*

$$R_{reg} - R_{reg}^* \lesssim (N\epsilon^2)^{-\frac{2\beta(p-1)}{2p\beta+d(p-1)}} + N^{-\frac{2\beta}{2\beta+d}}. \tag{26}$$

301   *Proof.* (Outline) It can be shown that the clipping bias scales as $T^{2(1-p)}$. To meet the $\epsilon$-label DP, an
302   additional error that scales as $T/\epsilon$ is needed. By averaging over $k$ nearest neighbors, the variance
303   caused by noise $W$ scales with $T^2/(k\epsilon^2)$. From standard analysis on nearest neighbor methods [29],
304   the non-private mean squared error scales as $1/k + (k/N)^{2\beta/d}$. Put all these terms together, Theorem
305   10 can be proved. Details can be found in Appendix K. $\qquad \square$

With the limit of $p \to \infty$, the problem reduces to the case with bounded noise, and the growth rate of $k$ and the convergence rate of risk are the same as those in Theorem 6. For finite $p$, $2\beta(p-1)/(2p\beta + d(p-1)) < 2\beta/(2\beta + d)$, thus the convergence rate becomes slower due to the privacy mechanism.

## 6.2 Central Label DP

*1) Lower bound.* The minimax lower bound is shown in Theorem 11.

**Theorem 11.** *The minimax lower bound is*

$$\inf_{\mathcal{A} \in \mathcal{A}_\epsilon} \sup_{(f,\eta) \in \mathcal{F}_{reg2}} (R_{reg} - R_{reg}^*) \gtrsim N^{-\frac{2\beta}{2\beta+d}} + (\epsilon N)^{-\frac{2\beta(p-1)}{p\beta+d(p-1)}} \tag{27}$$

*2) Upper bound.* Now we derive the upper bound. To restrict the sensitivity, instead of estimating with (23) directly, now we calculate an average of clipped label values:

$$\hat{\eta}_l = \frac{1}{n_l} \sum_{i=1}^{N} \mathbf{1}(\mathbf{X}_i \in B_l) \operatorname{Clip}(Y_i, T) + W_l, \tag{28}$$

in which $W_l \sim \operatorname{Lap}(2T/(n_l\epsilon))$. Then for all $\mathbf{x} \in B_l$, let $\hat{\eta}(\mathbf{x}) = \hat{\eta}_l$. The following theorem bounds the excess risk.

**Theorem 12.** (28) *is $\epsilon$-label DP. Moreover, under Assumption 1 and 3, if $h$ and $T$ scales as $h \sim N^{-\frac{1}{2\beta+d}} + (\epsilon N)^{-\frac{1}{p\beta+d(p-1)}}$, and $T \sim (\epsilon N h^d)^{1/p}$, then the excess risk can be bounded by*

$$R_{reg} - R_{reg}^* \lesssim N^{-\frac{2\beta}{2\beta+d}} + (\epsilon N)^{-\frac{2\beta(p-1)}{p\beta+d(p-1)}}. \tag{29}$$

The proof of Theorem 11 and 12 follow that of Theorem 7 and 8. The details are shown in Appendix L and M respectively. With $p = 2$, the right hand side of (29) becomes $(\epsilon \wedge 1)^{-\frac{2\beta}{2\beta+d}}$, indicating that the privacy constraint blows up the sample complexity by a constant factor. With larger $p$, the second term in (29) becomes negligible compared with the first one.

The theoretical analyses in this section are summarized as follows. In general, with fixed noise variance, if the label noise is heavy-tailed, while the non-private convergence rates remain unaffected, the additional risk caused by privacy mechanisms becomes significantly higher, indicating the difficulty of privacy protection for heavy-tailed distributions.

## 7 Conclusion

In this paper, we have derived the minimax lower bounds of learning under label DP for both central and local models. Furthermore, we propose methods whose upper bounds match these lower bounds. The results indicate the theoretical limits of learning under the label DP. From these results, it is discovered that under local label DP constraints, the sample complexity blows up by a factor of at least $O(1/\epsilon^2)$. Under central label DP requirements, the additional error caused by privacy mechanisms is significantly smaller. Finally, it is shown that for regression problem with heavy-tailed label distribution, the additional risk induced by privacy requirement becomes inevitably higher.

**Limitations:** The limitations of our work include the following aspects. Some assumptions can be weakened. For example, current analysis assumes that feature distributions have bounded supports, which may be extended to the unbounded case. One can let the bin splitting and nearest neighbor method be adaptive in the tails of features, such as [41]. Moreover, the bounds derived in this paper require that samples increase exponentially with dimensionality. However, in practice, the performance of learning under the label DP can be quite well even in high dimensions. The discrepancy can be explained by the fact that the minimax lower bound considers the worst-case distribution over a wide range of distributions. However, in most realistic cases, the distributions satisfy significantly better properties. A better modeling is to assume that these samples lie on a low dimensional manifold [57, 58]. In this case, it is possible to achieve a much better convergence rate. Finally, it is not sure whether approximate DP (i.e. $(\epsilon, \delta)$-DP) can improve the convergence rates.

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

# A  Proof of Proposition 2

478  From (5) and (6), the Bayes risk is

$$R_{cls}^* = \mathrm{P}(Y \neq c^*(\mathbf{X})) = \int \mathrm{P}(Y \neq c^*(\mathbf{x}) | \mathbf{X} = \mathbf{x}) f(\mathbf{x}) d\mathbf{x} = \int (1 - \eta^*(\mathbf{x})) f(\mathbf{x}) d\mathbf{x}. \tag{30}$$

479  The risk of classifier $c$ is

$$R_{cls} = \mathrm{P}(Y \neq c(\mathbf{X})) = \mathbb{E}\left[ \int \left(1 - \eta_{c(\mathbf{x})}(\mathbf{x})\right) f(\mathbf{x}) d\mathbf{x} \right]. \tag{31}$$

480  From (31) and (6),

$$R_{cls} - R_{cls}^* = \int (\eta^*(\mathbf{x}) - \mathbb{E}[\eta_{c(\mathbf{x})}(\mathbf{x})]) f(\mathbf{x}) d\mathbf{x}. \tag{32}$$

481  The proof is complete.

# B  Proof of Theorem 1

483  In this section, we prove the minimax lower bound of multi-class classification. The problem with $K$
484  classes with $K > 2$ is inherently harder than that with $K = 2$. Therefore, we just need to prove the
485  lower bound for binary classification, in which $\mathcal{Y} = \{1, 2\}$. Let

$$\eta(\mathbf{x}) = \eta_2(\mathbf{x}) - \eta_1(\mathbf{x}). \tag{33}$$

486  Since $\eta_1(\mathbf{x}) + \eta_2(\mathbf{x}) = 1$ always holds, we have

$$\eta_1(\mathbf{x}) = \frac{1 - \eta(\mathbf{x})}{2}, \eta_2(\mathbf{x}) = \frac{1 + \eta(\mathbf{x})}{2}. \tag{34}$$

487  Therefore, $\eta(\mathbf{x})$ captures the conditional distribution of $Y$ given $\mathbf{x}$.

488  Find $G$ disjoint cubes $B_1, \ldots, B_G \subset \mathcal{X}$, such that the length of each cube is $h$. Denote $\mathbf{c}_1, \ldots, \mathbf{c}_G$
489  as the centers of these cubes. Let $\phi(\mathbf{u})$ be some function supported at $[-1/2, 1/2]^d$, such that

$$0 \leq \phi(\mathbf{u}) \leq 1. \tag{35}$$

490  Let $f(\mathbf{x}) = c$ over $\mathbf{x} \in \mathcal{X}$. For $\mathbf{v} \in \mathcal{V} := \{-1, 1\}^m$, let

$$\eta_{\mathbf{v}}(\mathbf{x}) = \sum_{k=1}^m v_k \phi\left(\frac{\mathbf{x} - \mathbf{c}_k}{h}\right) h^\beta. \tag{36}$$

491  It can be proved that if for some constant $C_M$,

$$m \leq C_M h^{\gamma\beta - d}, \tag{37}$$

492  then for any $\eta = \eta_{\mathbf{v}}$, $\eta_1$ and $\eta_2$ satisfies Assumption 1(b). Denote

$$\hat{v}_k = \arg\max_{s \in \{-1,1\}} \int_{B_k} \phi\left(\frac{\mathbf{x} - \mathbf{c}_k}{h}\right) \mathbf{1}(\mathrm{sign}(\hat{\eta}(\mathbf{x})) = s) f(\mathbf{x}) d\mathbf{x}. \tag{38}$$

493  Then the excess risk is bounded by

$$\begin{aligned}
R - R^* &= \int |\eta_{\mathbf{v}}(\mathbf{x})| \mathrm{P}(\mathrm{sign}(\hat{\eta}(\mathbf{x})) \neq \mathrm{sign}(\eta_{\mathbf{v}}(\mathbf{x}))) f(\mathbf{x}) d\mathbf{x} \\
&\geq \sum_{k=1}^m \int_{B_k} |\eta_{\mathbf{v}}(\mathbf{x})| \mathrm{P}(\mathrm{sign}(\hat{\eta}(\mathbf{x})) \neq \mathrm{sign}(\eta_{\mathbf{v}}(\mathbf{x}))) f(\mathbf{x}) d\mathbf{x} \\
&= \sum_{k=1}^m h^\beta \int_{B_k} \phi\left(\frac{\mathbf{x} - \mathbf{c}_k}{h}\right) \mathrm{P}(\mathrm{sign}(\hat{\eta}(\mathbf{x}))) f(\mathbf{x}) d\mathbf{x}.
\end{aligned} \tag{39}$$

494  If $\hat{v}_k \neq v_k$, then from (38),

$$\int_{B_k} \phi\left(\frac{\mathbf{x} - \mathbf{c}_k}{h}\right) \mathbf{1}(\mathrm{sign}(\hat{\eta}(\mathbf{x}))) f(\mathbf{x}) d\mathbf{x} \geq \int_{B_k} \phi\left(\frac{\mathbf{x} - \mathbf{c}_k}{h}\right) \mathbf{1}(\mathrm{sign}(\hat{\eta}(\mathbf{x})) = v_k) f(\mathbf{x}) d\mathbf{x}. \tag{40}$$

Therefore

$$\int_{B_k} \phi\left(\frac{\mathbf{x}-\mathbf{c}_k}{h}\right) \mathbf{1}(\text{sign}(\hat{\eta}(\mathbf{x})) \neq v_k) f(\mathbf{x}) d\mathbf{x} \geq \frac{1}{2} \int_{B_k} \phi\left(\frac{\mathbf{x}-\mathbf{c}_k}{h}\right) f(\mathbf{x}) d\mathbf{x} \geq \frac{1}{2} ch^d \|\phi\|_1. \quad (41)$$

Hence

$$\begin{aligned} R - R^* &\geq \frac{1}{2} ch^{\beta+d} \|\phi\|_1 \sum_{k=1}^m \mathbf{P}(\hat{v}_k \neq v_k) \\ &= \frac{1}{2} ch^{\beta+d} \|\phi\|_1 \mathbb{E}[\rho_H(\hat{\mathbf{v}}, \mathbf{v})], \end{aligned} \quad (42)$$

in which $\rho_H$ denotes the Hamming distance. Then

$$\inf_{\hat{Y}} \inf_{M \in \mathcal{M}_\epsilon} \sup_{(f,\eta) \in \mathcal{P}} (R - R^*) \geq \frac{1}{2} h^{\beta+d} \|\phi\|_1 \inf_{\hat{\mathbf{v}}} \inf_{M \in \mathcal{M}_\epsilon} \max_{\mathbf{v} \in \mathcal{V}} \mathbb{E}[\rho_H(\hat{\mathbf{v}}, \mathbf{v})]. \quad (43)$$

Define

$$\delta = \sup_{M \in \mathcal{M}_\epsilon} \max_{\mathbf{v},\mathbf{v}':\rho_H(\mathbf{v},\mathbf{v}')=1} D_{KL}(P_{(X,Z)_{1:N}|\mathbf{v}} \| P_{(X,Z)_{1:N}|\mathbf{v}'}), \quad (44)$$

in which $P_{(X,Z)_{1:N}|\mathbf{v}}$ denotes the distribution of $(\mathbf{X}_1, Z_1), \ldots, (\mathbf{X}_N, Z_N)$ with $\eta = \eta_{\mathbf{v}}$. $D_{KL}$ denotes the Kullback-Leibler divergence. Then from [28], Theorem 2.12(iv),

$$\inf_{\hat{\mathbf{v}}} \inf_M \max_{\mathbf{v} \in \mathcal{V}} \mathbb{E}[\rho_H(\hat{\mathbf{v}}, \mathbf{v})] \geq \frac{m}{2}\left(\frac{1}{2}e^{-\delta}, 1 - \sqrt{\frac{\delta}{2}}\right). \quad (45)$$

It remains to bound $\delta$. Without loss of generality, suppose $v_1 \neq v_1'$, and $v_i = v_i'$ for $i \neq 1$. Then

$$\begin{aligned} D_{KL}(P_{(X,Z)_{1:N}|\mathbf{v}} \| P_{(X,Z)_{1:N}|\mathbf{v}'}) &\overset{(a)}{=} N D_{KL}(P_{X,Z|\mathbf{v}} \| P_{X,Z|\mathbf{v}'}) \\ &\overset{(b)}{=} N \int_{B_1} f(\mathbf{x}) D_{KL}(P_{Z|\mathbf{X}=\mathbf{x},\mathbf{v}} \| P_{Z|\mathbf{X}=\mathbf{x},\mathbf{v}'}) d\mathbf{x} \\ &\overset{(c)}{\leq} N \int_{B_1} f(\mathbf{x})(e^\epsilon - 1)^2 \mathbb{TV}^2(P_{Z|\mathbf{X}=\mathbf{x},\mathbf{v}}, P_{Z|\mathbf{X}=\mathbf{x},\mathbf{v}'}) d\mathbf{x} \\ &= N \int_{B_1} f(\mathbf{x})(e^\epsilon - 1)^2 \eta_{\mathbf{v}}^2(\mathbf{x}) d\mathbf{x} \\ &= N(e^\epsilon - 1)^2 \int_{B_1} f(\mathbf{x})\phi^2\left(\frac{\mathbf{x}-\mathbf{c}_1}{h}\right) h^{2\beta} d\mathbf{x} \\ &\overset{(d)}{=} N(e^\epsilon - 1)^2 h^{2\beta+d} \|\phi\|_2^2. \end{aligned} \quad (46)$$

In (a), $P_{X,Z|\mathbf{v}}$ denotes the distribution of a single sample with privatized label $(X, Z)$, with $\eta = \eta_{\mathbf{v}}$. In (b), $P_{Z|\mathbf{X}=\mathbf{x},\mathbf{v}}$ denotes the conditional distribution of $Z$ given $\mathbf{X} = \mathbf{x}$, with $\eta = \eta_{\mathbf{v}}$. (c) uses [24], Theorem 1. In (d), $\|\phi\|_2^2 = \int \phi^2(\mathbf{u}) d\mathbf{u}$, which is a constant. Moreover,

$$\begin{aligned} D_{KL}(P_{X,Z|\mathbf{v}} \| P_{X,Z|\mathbf{v}'}) &\overset{(a)}{\leq} D_{KL}(P_{X,Y|\mathbf{v}} \| P_{X,Y|\mathbf{v}'}) \\ &= \int_{B_1} f(\mathbf{x})\left[\mathbf{P}(Y=1|\mathbf{v})\ln\frac{\mathbf{P}(Y=1|\mathbf{v})}{\mathbf{P}(Y=1|\mathbf{v}')} + \mathbf{P}(Y=-1|\mathbf{v})\ln\frac{\mathbf{P}(Y=-1|\mathbf{v})}{\mathbf{P}(Y=-1|\mathbf{v}')}\right] d\mathbf{x} \\ &= \int_{B_1} f(\mathbf{x})\left[\frac{1+\eta_{\mathbf{v}}(\mathbf{x})}{2}\ln\frac{1+\eta_{\mathbf{v}}(\mathbf{x})}{1-\eta_{\mathbf{v}}(\mathbf{x})} + \frac{1-\eta_{\mathbf{v}}(\mathbf{x})}{2}\ln\frac{1-\eta_{\mathbf{v}}(\mathbf{x})}{1+\eta_{\mathbf{v}}(\mathbf{x})}\right] d\mathbf{x} \\ &\overset{(b)}{\leq} 3 \int_{B_1} f(\mathbf{x})\eta_{\mathbf{v}}^2(\mathbf{x}) d\mathbf{x} \\ &\leq 3h^{2\beta+d} \|\phi\|_2^2. \end{aligned} \quad (47)$$

For (a), note that $Z$ is generated from $Y$. From data processing inequality, (a) holds. For (b), without loss of generality, suppose that $v_1 = 1$, thus $\eta_{\mathbf{v}}(\mathbf{x}) \geq 0$ in $B_1$. Then $\ln(1 + \eta_{\mathbf{v}}(\mathbf{x})) \leq \eta_{\mathbf{v}}(\mathbf{x})$. From (35) and (36), $|\eta_{\mathbf{v}}(\mathbf{x})| \leq 1/2$. Therefore, $-\ln(1 - \eta_{\mathbf{v}}(\mathbf{x})) \leq 2\eta_{\mathbf{v}}(\mathbf{x})$. Therefore (b) holds.

From (46) and (47),

$$\delta \leq N \left[ (e^\epsilon - 1)^2 \wedge 3 \right] h^{2\beta+d} \|\phi\|_2^2. \tag{48}$$

Let

$$h \sim \left( N \left( \epsilon^2 \wedge 1 \right) \right)^{-\frac{1}{2\beta+d}}. \tag{49}$$

Then $\delta \lesssim 1$. From (45), with $m \sim h^{\gamma\beta-d}$,

$$\inf_{\hat{\mathbf{v}}} \inf_{M \in \mathcal{M}_\epsilon} \max_{\mathbf{v} \in \mathcal{V}} \mathbb{E}[\rho_H(\hat{\mathbf{v}}, \mathbf{v})] \gtrsim h^{\gamma\beta-d}. \tag{50}$$

Hence

$$\inf_{\hat{Y}} \inf_{M \in \mathcal{M}_\epsilon} \sup_{(f,\eta) \in \mathcal{P}} (R - R^*) \gtrsim h^{\beta+d} h^{\gamma\beta-d} \sim h^{\beta(\gamma+1)} \sim \left[ N \left( \epsilon^2 \wedge 1 \right) \right]^{-\frac{\beta(\gamma+1)}{2\beta+d}}. \tag{51}$$

The proof is complete.

# C   Proof of Theorem 2

Denote

$$n_l = \sum_{i=1}^{N} \mathbf{1}(\mathbf{X}_i \in B_l), \tag{52}$$

and for $\mathbf{Z} = M(\mathbf{X}, Y)$, let

$$\begin{aligned} \tilde{\eta}_j(\mathbf{x}) &:= \mathbb{E}[\mathbf{Z}(j)|\mathbf{X} = \mathbf{x}] \\ &= \frac{e^{\frac{\epsilon}{2}}}{e^{\frac{\epsilon}{2}} + 1} \eta_j(\mathbf{x}) + \frac{1}{e^{\frac{\epsilon}{2}} + 1} (1 - \eta_j(\mathbf{x})) \end{aligned} \tag{53}$$

as the number of training samples whose feature vectors fall in $B_l$, and

$$v_{lj} := \frac{1}{n_l} \sum_{i:\mathbf{X}_i \in B_l} \tilde{\eta}_j(\mathbf{X}_i). \tag{54}$$

Recall (12) that defines $S_{lj}$. From Hoeffding's inequality,

$$\mathrm{P}\left(|S_{lj} - n_l v_{lj}| > t | \mathbf{X}_{1:N}\right) \leq 2 \exp\left[-\frac{2t^2}{n_l}\right], \tag{55}$$

in which $\mathbf{X}_{1:N}$ denotes $\mathbf{X}_1, \ldots, \mathbf{X}_N$.

Define

$$v_l^* := \max_j v_{lj}, \tag{56}$$

and

$$c_l^* := \arg\max_j v_{lj}. \tag{57}$$

Now we bound $\mathrm{P}(v_l^* - v_{lc_l} > t)$, in which $c_l$ is defined in (13). $c_l$ can be viewed as the prediction at the $l$-th bin. We would like to show that the even if the prediction is wrong, the value (i.e. conditional probability) of the predicted class is close to the ground truth. $v_l^* - v_{lc_l} > t$ only if $\exists j$, $v_l^* - v_{lj} > t$, and $S_{lj} > S_{lc_l^*}$. Therefore either $S_{lj} - n_l v_{lj} > t/2$ or $S_{lc_l^*} - n_l v_l^* > t/2$ holds. Hence

$$\mathrm{P}\left(v_l^* - v_{lc_l} \geq t\right) \leq \mathrm{P}\left(\exists j, |S_{lj} - n_l v_{lj}| \geq \frac{1}{2} n_l t\right) \leq 2K \exp\left(-\frac{1}{2} n_l t^2\right). \tag{58}$$

Define

$$t_0 = \sqrt{\frac{2\ln(2K)}{n_l}}. \tag{59}$$

526 Then

$$
\begin{aligned}
v_l^* - \mathbb{E}[v_{lc_l}|\mathbf{X}_{1:N}] \quad &= \quad \int_0^1 \mathbf{P}(v_l^* - v_{lc_l} > t)dt \\
&\leq \quad t_0 + \int_{t_0}^{\infty} 2K \exp\left(-\frac{1}{2}n_l t^2\right) dt \\
&\overset{(a)}{\leq} \quad t_0 + 2\sqrt{\frac{2\pi}{n_l}} K \exp\left(-\frac{1}{2}n_l t_0^2\right) \\
&= \quad \sqrt{\frac{2\ln(2K)}{n_l}} + \sqrt{\frac{2\pi}{n_l}} \\
&\leq \quad 3\sqrt{\frac{\ln(2K)}{n_l}}.
\end{aligned}
\tag{60}
$$

527 In (a), we use the inequality

$$
\int_t^{\infty} e^{-\frac{u^2}{2\sigma^2}} du \leq \sqrt{2\pi}\sigma e^{-\frac{t^2}{2\sigma^2}}.
\tag{61}
$$

528 Now we bound the excess risk.

$$
\begin{aligned}
R - R^* \quad &= \quad \int \left(\eta^*(\mathbf{x}) - \mathbb{E}[\eta_{c(\mathbf{x})}(\mathbf{x})]\right) f(\mathbf{x})d\mathbf{x} \\
&= \quad \sum_{l=1}^{G} \int_{B_l} \left(\eta^*(\mathbf{x}) - \mathbb{E}[\eta_{c(\mathbf{x})}(\mathbf{x})]\right) f(\mathbf{x})d\mathbf{x}.
\end{aligned}
\tag{62}
$$

529 We need to bound $\int_{B_l} \left(\eta^*(\mathbf{x}) - \mathbb{E}[\eta_{c(\mathbf{x})}(\mathbf{x})]\right) f(\mathbf{x})d\mathbf{x}$ for each $l$. From Assumption 1(a), for any
530 $\mathbf{x}, \mathbf{x}' \in B_l$, the distance is bounded by $\|\mathbf{x} - \mathbf{x}'\| \leq \sqrt{d}L$. Thus

$$
|\eta_j(\mathbf{x}) - \eta_j(\mathbf{x}')| \leq L_d h^\beta,
\tag{63}
$$

531 in which $L_d$ is defined as $L_d := L\sqrt{d}$. From (63) and (53),

$$
|\tilde{\eta}_j(\mathbf{x}) - \tilde{\eta}_j(\mathbf{x}')| \leq \frac{e^{\frac{\epsilon}{2}} - 1}{e^{\frac{\epsilon}{2}} + 1} L_d h^\beta.
\tag{64}
$$

532 Define

$$
\tilde{\eta}^*(\mathbf{x}) = \max_j \tilde{\eta}_j(\mathbf{x}),
\tag{65}
$$

533 then

$$
\begin{aligned}
\eta^*(\mathbf{x}) - \mathbb{E}[\eta_{c_l}(\mathbf{x})|\mathbf{X}_{1:N}] \quad &\leq \quad \frac{e^{\frac{\epsilon}{2}} + 1}{e^{\frac{\epsilon}{2}} - 1} \left(\tilde{\eta}^*(\mathbf{x}) - \mathbb{E}[\tilde{\eta}_{c_l}(\mathbf{x})|\mathbf{X}_{1:N}]\right) \\
&\leq \quad \frac{e^{\frac{\epsilon}{2}} + 1}{e^{\frac{\epsilon}{2}} - 1} \left(v_l^* - \mathbb{E}[v_{lc_l}|\mathbf{X}_{1:N}]\right) + 2L_d h^\beta \\
&\leq \quad 3\frac{e^{\frac{\epsilon}{2}} + 1}{e^{\frac{\epsilon}{2}} - 1}\sqrt{\frac{2\ln(2K)}{n_l}} + 2L_d h^\beta.
\end{aligned}
\tag{66}
$$

534 Take integration over cube $B_l$, we get

$$
\begin{aligned}
&\int_{B_l} \left(\eta^*(\mathbf{x}) - \mathbb{E}[\eta_{c_l}(\mathbf{x})]\right) f(\mathbf{x})d\mathbf{x} \\
&\leq \quad \mathbf{P}\left(n_l < \frac{1}{2}Np(B_l)\right) \int_{B_l} \left(\eta^*(\mathbf{x}) - \mathbb{E}[\eta_{c_l}(\mathbf{x})|n_l < \frac{1}{N}p(B_l)]\right) f(\mathbf{x})d\mathbf{x} \\
&\quad + \int_{B_l} \left(\eta^*(\mathbf{x}) - \mathbb{E}[\eta_{c_l}(\mathbf{x})|n_l \geq \frac{1}{N}p(B_l)]\right) f(\mathbf{x})d\mathbf{x} \\
&\leq \quad p(B_l)e^{-\frac{1}{2}(1-\ln 2)Np(B_l)} + \left[3\frac{e^{\frac{\epsilon}{2}} + 1}{e^{\frac{\epsilon}{2}} - 1}\sqrt{\frac{2\ln(2K)}{Np(B_l)}} + 2L^d h^\beta\right] p(B_l),
\end{aligned}
\tag{67}
$$

in which $p(B_l) = P(\mathbf{X} \in B_l)$ is the probability mass of $B_l$. Moreover, define

$$\Delta_l = \inf_{\mathbf{x} \in B_l} \left( \eta^*(\mathbf{x}) - \eta_s(\mathbf{x}) \right), \tag{68}$$

and

$$\tilde{\Delta}_l = \inf_{\mathbf{x} \in B_l} \left( \tilde{\eta}^*(\mathbf{x}) - \tilde{\eta}_s(\mathbf{x}) \right) = \frac{e^{\frac{\epsilon}{2}} - 1}{e^{\frac{\epsilon}{2}} + 1} \Delta_l, \tag{69}$$

in which the $\tilde{\eta}_s$ is the second largest value of $\tilde{\eta}_j$ among $j = 1, \ldots, K$, which follows the definition of $\eta_s$.

If $\Delta_l > 0$, then $c^*(\mathbf{x})$ is the same over $B_l$. Then either $v_l^* - v_{lc_l} = 0$ or $v_l^* - v_{lc_l} \geq \Delta_l$ holds. Hence

$$
\begin{aligned}
& \tilde{\eta}^*(\mathbf{x}) - \mathbb{E}[\tilde{\eta}_{c_l}(\mathbf{x})|\mathbf{X}_{1:N}] \\
&= \int_0^1 P\left( \tilde{\eta}^*(\mathbf{x}) - \tilde{\eta}_{c_l}(\mathbf{x}) > t | \mathbf{X}_{1:N} \right) dt \\
&\leq \int_0^1 P\left( v_l^* - v_{lc_l} > t - 2L_d h^\beta \frac{e^{\frac{\epsilon}{2}} + 1}{e^{\frac{\epsilon}{2}} - 1} | \mathbf{X}_{1:N} \right) dt \\
&\leq \int_0^{\tilde{\Delta}_l + 2L_d h^\beta} P(v_l^* - v_{lc_l} \geq \Delta_l) dt + \int_{\tilde{\Delta}_l + 2L_d h^\beta}^\infty 2K \exp\left[ -\frac{1}{2} n_l (t - 2L_d h^\beta)^2 \right] dt \\
&\leq 2K \exp\left( -\frac{1}{2} n_l \tilde{\Delta}_l^2 \right) (\tilde{\Delta}_l + 2L_d h^\beta \frac{e^{\frac{\epsilon}{2}} + 1}{e^{\frac{\epsilon}{2}} - 1}) + 2K \sqrt{\frac{2\pi}{n_l}} \exp\left( -\frac{1}{2} n_l \tilde{\Delta}_l^2 \right) \\
&= \left[ 2K \left( \tilde{\Delta}_l + 2L_d h^\beta \frac{e^{\frac{\epsilon}{2}} + 1}{e^{\frac{\epsilon}{2}} - 1} \right) + 2K \sqrt{\frac{2\pi}{n_l}} \right] \exp\left( -\frac{1}{2} n_l \tilde{\Delta}_l^2 \right).
\end{aligned}
\tag{70}
$$

Take expectation over $\mathbf{X}_{1:N}$, we get

$$
\begin{aligned}
\int_{B_l} (\eta^*(\mathbf{x}) - \mathbb{E}[\eta_{c_l}(\mathbf{x})]) f(\mathbf{x}) d\mathbf{x} \leq p(B_l) e^{-\frac{1}{2}(1 - \ln 2) N p(B_l)} \\
+ 2K p(B_l) \left( \Delta_l + 2L_d h^\beta + \frac{e^{\frac{\epsilon}{2}} + 1}{e^{\frac{\epsilon}{2}} - 1} \sqrt{\frac{2\pi}{N p(B_l)}} \right) \exp\left[ -\frac{1}{2} N p(B_l) \Delta_l^2 \left( \frac{e^{\frac{\epsilon}{2}} - 1}{e^{\frac{\epsilon}{2}} + 1} \right)^2 \right]
\end{aligned}
\tag{71}
$$

Define

$$a_l = \left[ 3 \frac{e^{\frac{\epsilon}{2}} + 1}{e^{\frac{\epsilon}{2}} - 1} \sqrt{\frac{2 \ln(2K)}{cNh^d}} + 2L_d h^\beta \right] p(B_l), \tag{72}$$

and

$$b_l = 2K p(B_l) \left( \Delta_l + 2L_d h^\beta + \frac{e^{\frac{\epsilon}{2}} + 1}{e^{\frac{\epsilon}{2}} - 1} \sqrt{\frac{2\pi}{cNh^d}} \right) \exp\left[ -\frac{1}{2} cNh^d \Delta_l^2 \left( \frac{e^{\frac{\epsilon}{2}} - 1}{e^{\frac{\epsilon}{2}} + 1} \right)^2 \right]. \tag{73}$$

From Assumption 1(c), $p(B_l) \geq cNh^d$. Therefore, from (67) and (71)

$$
\begin{aligned}
R - R^* &\leq \sum_{l=1}^G \left[ p(B_l) e^{-\frac{1}{2}(1 - \ln 2) N p(B_l)} + \min\{a_l, b_l\} \right] \\
&\leq e^{-\frac{1}{2}(1 - \ln 2) cNh^d} + \sum_{l=1}^G \min\{a_l, b_l\}.
\end{aligned}
\tag{74}
$$

It remains to bound $\sum_{l=1}^G \min\{a_l, b_l\}$. Note that for all $\mathbf{x} \in B_l$, $\eta^*(\mathbf{x}) - \eta_s(\mathbf{x}) \leq \Delta_l + 2L_d h^\beta$. Thus

$$\sum_{l:\Delta_l \leq u} p(B_l) \leq P\left( \eta^*(\mathbf{X}) - \eta_s(\mathbf{X}) \leq u + 2L_d h^\beta \right) \leq M(u + 2L_d h^\beta)^\gamma. \tag{75}$$

Let

$$\Delta_0 = \frac{e^{\frac{\epsilon}{2}} + 1}{e^{\frac{\epsilon}{2}} - 1} \sqrt{\frac{2 \ln(2K)}{cNh^d}}, \tag{76}$$

and

$$I_0 = \{l | \Delta_l \le \Delta_0\}, \tag{77}$$
$$I_k = \{l | 2^{k-1} \Delta_0 < \Delta_l \le 2^k \Delta_0\}, k = 1, 2, \dots \tag{78}$$

Then

$$
\begin{aligned}
\min_{l \in I_0} \{a_l, b_l\} &\le \sum_{l \in I_0} a_l \\
&\le \left( \sum_{l : \Delta_l \le \Delta_0} p(B_l) \right) \left[ 3 \frac{e^{\frac{\epsilon}{2}} + 1}{e^{\frac{\epsilon}{2}} - 1} \sqrt{\frac{2 \ln(2K)}{cNh^d}} + 2L_d h^\beta \right] \\
&\le M(\Delta_0 + 2L_d h^\beta)^\gamma \left[ 3 \frac{e^{\frac{\epsilon}{2}} + 1}{e^{\frac{\epsilon}{2}} - 1} \sqrt{\frac{2 \ln(2K)}{cNh^d}} + 2L_d h^\beta \right] \\
&\lesssim \left( \frac{1}{\epsilon^2 \wedge 1} \frac{\ln K}{Nh^d} \right)^{\frac{\gamma+1}{2}} + h^{\beta(\gamma+1)}. \tag{79}
\end{aligned}
$$

For $I_k$ with $k \ge 1$,

$$
\begin{aligned}
\min_{l \in I_k} \{a_l, b_l\} &\le \sum_{l \in I_k} b_l \\
&\le \left( \sum_{l : \Delta_l \le 2^k \Delta_0} p(B_l) \right) \cdot 2K \left( 2^k \Delta_0 + 2L_d h^\beta + \Delta_0 \right) \exp \left[ -\frac{1}{2} \left( \frac{e^{\frac{\epsilon}{2}} - 1}{e^{\frac{\epsilon}{2}} + 1} \right)^2 cNh^d 2^{2k-2} \Delta_0^2 \right] \\
&\le M(2^k \Delta_0 + 2L_d h^\beta)^\gamma \left( (2^k + 1) \Delta_0 + 2L_d h^\beta \right) (2K)^{-2^{2k-2}+1} \\
&\le M(\Delta_0 + 2L_d h^\beta)^{\gamma+1} 2^{k\gamma + k - 2^{2k-2}+2}. \tag{80}
\end{aligned}
$$

It is obvious that there exists a finite constant $C' < \infty$ that depends on $\gamma$, such that

$$\sum_{k=1}^\infty 2^{k\gamma + k - 2^{2k-2}+2} \le C'. \tag{81}$$

Therefore

$$\sum_{k=1}^\infty \sum_{l \in I_k} \min\{a_l, b_l\} \lesssim \left( \frac{1}{\epsilon^2 \wedge 1} \frac{\ln K}{Nh^d} \right)^{\frac{\gamma+1}{2}} + h^{\beta(\gamma+1)}. \tag{82}$$

Combine (74), (79) and (82),

$$R - R^* \lesssim \left( \frac{1}{\epsilon^2 \wedge 1} \frac{\ln K}{Nh^d} \right)^{\frac{\gamma+1}{2}} + h^{\beta(\gamma+1)}. \tag{83}$$

To minimize the overall excess risk, let

$$h \sim \left( \frac{N(\epsilon^2 \wedge 1)}{\ln K} \right)^{-\frac{1}{2\beta+d}}, \tag{84}$$

then

$$R - R^* \lesssim \left( \frac{N(\epsilon^2 \wedge 1)}{\ln K} \right)^{-\frac{\beta(\gamma+1)}{2\beta+d}}. \tag{85}$$

Compare to the simple random response method, the bin splitting avoids the polynomial decrease over $K$.

## D  Proof of Theorem 3

We still divide the support as the local label DP setting, except that the value of $h$ is different, which will be specified later in this section. Note that (42) still holds here. Let $\mathbf{V}$ takes values from $\{-1, 1\}^m$ randomly with equal probability, and $V_k$ is the $k$-th element. Then $\eta_{\mathbf{V}}(\mathbf{x})$ is a random function. The corresponding random output of hypothesis testing is denoted as $\hat{V}_k$, which is calculated by (38). Then

$$
\begin{aligned}
\inf_{\mathcal{A} \in \mathcal{A}_\epsilon} \sup_{(f,\eta) \in \mathcal{F}_{cls}} (R - R^*) &\geq \frac{1}{2} ch^{\beta+d} \|\phi\|_1 \inf_{\mathcal{A} \in \mathcal{A}_\epsilon} \max_{\mathbf{v} \in \mathcal{V}} \sum_{k=1}^{m} \mathrm{P}(\hat{v}_k \neq v_k) \\
&\geq \frac{1}{2} h^{\beta+d} \|\phi\|_1 \inf_{\mathcal{A} \in \mathcal{A}_\epsilon} \sum_{k=1}^{m} \mathrm{P}(\hat{V}_k \neq V_k) \\
&= \frac{1}{2} h^{\beta+d} \|\phi\|_1 \sum_{k=1}^{m} \inf_{\mathcal{A} \in \mathcal{A}_\epsilon} \mathrm{P}(\hat{V}_k \neq V_k),
\end{aligned}
\tag{86}
$$

in which the last step holds since $\hat{V}_k$ for different $k$ are calculated independently.

It remains to give a lower bound of $\mathrm{P}(\hat{V}_k \neq V_k)$. Denote $n_k$ as the number of samples falling in $B_k$, $\bar{Y}_k$ as the average label values in $B_k$:

$$
n_k := \sum_{i=1}^{N} \mathbf{1}(\mathbf{X}_i \in B_k),
\tag{87}
$$

$$
\bar{Y}_k := \frac{1}{n_k} \sum_{i=1}^{N} Y_i \mathbf{1}(X_i \in B_k).
\tag{88}
$$

Moreover, define

$$
\begin{aligned}
a_k &:= \frac{1}{n_k} \sum_{i=1}^{N} |\eta(\mathbf{X}_i)| \mathbf{1}(\mathbf{X}_i \in B_k) \\
&= \frac{h^\beta}{n_k} \sum_{i=1}^{N} \phi\left(\frac{\mathbf{X}_i - \mathbf{c}_k}{h}\right) \mathbf{1}(\mathbf{X}_i \in B_k),
\end{aligned}
\tag{89}
$$

in which the last step comes from (36). Then

$$
\mathbb{E}[\bar{Y}_k | \mathbf{X}_{1:N}, V_k] = V_k a_k,
\tag{90}
$$

in which $\mathbf{X}_{1:N}$ means $\mathbf{X}_1, \ldots, \mathbf{X}_N$. We then show the following lemma.

**Lemma 1.** *If* $0 \leq t \leq \ln 2/(\epsilon n_k)$, *and* $n_k t$ *is an integer, then*

$$
P(\hat{V}_k = 1 | \mathbf{X}_{1:N}, \bar{Y}_k = -t) + P(\hat{V}_k = -1 | \mathbf{X}_{1:N}, \bar{Y}_k = t) \geq \frac{2}{3}.
\tag{91}
$$

*Proof.* Construct $D'$ by changing the label values of $l = n_k t$ items from these $n_k$ samples falling in $B_k$, from $-1$ to $1$. Then the average label values in $B_k$ is denoted as $\bar{Y}_k'$ after such replacement. $\hat{V}_k$ also becomes $\hat{V}_k'$. Then from the $\epsilon$-label DP requirement,

$$
\begin{aligned}
\mathrm{P}(\hat{V}_k = 1 | \mathbf{X}_{1:N}, \bar{Y}_k = -t) &\overset{(a)}{\geq} e^{-l\epsilon} \mathrm{P}\left(\hat{V}_k' = 1 | \mathbf{X}_{1:N}, \bar{Y}_k' = -t + \frac{2l}{n_k}\right) \\
&\overset{(b)}{\geq} e^{-l\epsilon} \mathrm{P}\left(\hat{V}_k = 1 | \mathbf{X}_{1:N}, \bar{Y}_k = -t + \frac{2l}{n_k}\right) \\
&\geq e^{-n_k t\epsilon} \left[1 - \mathrm{P}\left(\hat{V}_k = -1 | \mathbf{X}_{1:N}, \bar{Y}_k = -t + \frac{2l}{n_k}\right)\right] \\
&\geq \frac{1}{2} \left[1 - \mathrm{P}\left(\hat{V}_k = -1 | \mathbf{X}_{1:N}, \bar{Y}_k = t\right)\right].
\end{aligned}
\tag{92}
$$

in which (a) uses the group privacy property. The Hamming distance between $D$ and $D'$ is $l$, thus the ratio of probability between $D$ and $D'$ is within $[e^{-l\epsilon}, e^{l\epsilon}]$. (b) holds because the algorithm does not change after changing $D$ to $D'$. Similarly,

$$P(\hat{V}_k = -1|\mathbf{X}_{1:N}, \bar{Y}_k = t) \geq \frac{1}{2}\left[1 - P\left(\hat{V}_k = 1|\mathbf{X}_{1:N}, \bar{Y}_k = -t\right)\right]. \tag{93}$$

Then (91) can be shown by adding up (92) and (93). $\qquad\qquad\square$

Now we use Lemma 1 to bound the excess risk. With sufficiently large $n_k$, $\hat{Y}_k$ will be close to Gaussian distribution with mean $a_k$. To be more rigorous, by Berry-Esseen theorem [?], for some absolute constant $C_E$,

$$P\left(\bar{Y}_k \leq a_k|\mathbf{X}_{1:N}, V_k = 1\right) \geq \frac{1}{2} - \frac{C_E}{\sqrt{n_k}}. \tag{94}$$

Similarly,

$$P\left(\bar{Y}_k \geq -a_k|\mathbf{X}_{1:N}, V_k = -1\right) \geq \frac{1}{2} - \frac{C_E}{\sqrt{n_k}}. \tag{95}$$

We first analyze cubes with

$$n_k > 16C_E^2, a_k < \frac{\ln 2}{\epsilon n_k}. \tag{96}$$

Under condition (96), the right hand side of (94) and (95) are at least $1/4$. Therefore

$$
\begin{aligned}
P(\hat{V}_k \neq V_k|\mathbf{X}_{1:N}) &= \frac{1}{2}P(\hat{V}_k = 1|\mathbf{X}_{1:N}, V_k = -1) + \frac{1}{2}P(\hat{V}_k = -1|\mathbf{X}_{1:N}, V_k = 1) \\
&\geq \frac{1}{8}P\left(\hat{V}_k = 1|\mathbf{X}_{1:N}, \bar{Y}_k \geq -\frac{\ln 2}{\epsilon n_k}\right) + \frac{1}{8}P\left(\hat{V}_k = -1|\mathbf{X}_{1:N}, \bar{Y}_k \leq \frac{\ln 2}{\epsilon n_k}\right) \\
&\geq \frac{1}{12}.
\end{aligned} \tag{97}
$$

From (86),

$$\inf_{\mathcal{A}\in\mathcal{A}_\epsilon}\sup_{(f,\eta)\in\mathcal{F}_{cls}} (R - R^*) \geq \frac{1}{2}h^{\beta+d}\|\phi\|_1 \sum_{k=1}^{m}\frac{1}{12}P\left(a_k < \frac{\ln 2}{\epsilon n_k}, n_k > 16C_E^2\right) \tag{98}$$

From (35), (89) and (87), $a_k \leq h^\beta$. Therefore

$$\inf_{\mathcal{A}\in\mathcal{A}_\epsilon}\sup_{(f,\eta)\in\mathcal{F}_{cls}} (R - R^*) \geq \frac{1}{24}h^{\beta+d}\|\phi\|_1 \sum_{k=1}^{m}P\left(16C_E^2 < n_k < \frac{\ln 2}{\epsilon h^\beta}\right). \tag{99}$$

Recall that each cube has probability mass $ch^d$. Select $h$ such that

$$2Nch^d = \frac{\ln 2}{\epsilon h^\beta}. \tag{100}$$

From Chernoff inequality, $16C_E^2 < n_k < \ln 2/(\epsilon h^\beta)$ holds with high probability. (100) yields

$$h \sim (\epsilon N)^{-\frac{1}{d+\beta}}. \tag{101}$$

Recall the bound of $m$ in (37). Let $m \sim h^{\gamma\beta-d}$, then (99) becomes

$$
\begin{aligned}
\inf_{\mathcal{A}\in\mathcal{A}_\epsilon}\sup_{(f,\eta)\in\mathcal{F}_{cls}} (R - R^*) &\gtrsim h^{\beta(\gamma+1)} \\
&\gtrsim (\epsilon N)^{-\frac{\beta(\gamma+1)}{d+\beta}}.
\end{aligned} \tag{102}
$$

Moreover, the standard lower bound for classification [28] is

$$\inf_{\mathcal{A}\in\mathcal{A}_\epsilon}\sup_{(f,\eta)\in\mathcal{F}_{cls}} (R - R^*) \gtrsim N^{-\frac{\beta(\gamma+1)}{2\beta+d}}. \tag{103}$$

Therefore

$$\inf_{\mathcal{A}\in\mathcal{A}_\epsilon}\sup_{(f,\eta)\in\mathcal{F}_{cls}} (R - R^*) \gtrsim N^{-\frac{\beta(\gamma+1)}{2\beta+d}} + (\epsilon N)^{-\frac{\beta(\gamma+1)}{d+\beta}}. \tag{104}$$

 # E   Proof of Theorem 4

591   Denote
$$n_l^* = \max_j n_{lj}, \tag{105}$$

592
$$n_l := \sum_{j=1}^{K} n_{lj} = \sum_{i=1}^{N} \mathbf{1}(\mathbf{X}_i \in B_l). \tag{106}$$

593   For all $j$ such that $n_l^* - n_{lj} > t$,
$$\begin{aligned} \mathrm{P}(c_l = j | \mathbf{X}_{1:N}, Y_{1:N}) &= \frac{e^{\epsilon n_{lj}/2}}{\sum_{k=1}^{K} e^{\epsilon n_{lk}/2}} \\ &\leq \frac{e^{\epsilon n_l^*/2}}{\sum_{k=1}^{K} e^{\epsilon n_{lk}/2}} e^{-\frac{1}{2}\epsilon t} \\ &\leq e^{-\frac{1}{2}\epsilon t}. \end{aligned} \tag{107}$$

594   Therefore
$$\mathrm{P}(n_l^* - n_{lc_l} > t) = \sum_{j:n_l^* - n_{lj} > t} \mathrm{P}(c_l = j | \mathbf{X}_{1:N}, Y_{1:N}) \leq K e^{-\frac{1}{2}\epsilon t}. \tag{108}$$

595   Hence
$$\begin{aligned} \mathbb{E}[n_l^* - n_{lc_l}] &= \int_0^\infty \mathrm{P}(n_l^* - n_{lj} > t) dt \\ &\leq \int_0^{2\ln K/\epsilon} 1 dt + \int_{2\ln K/\epsilon}^\infty K e^{-\frac{1}{2}\epsilon t} dt \\ &= \frac{2}{\epsilon}(\ln K + 1). \end{aligned} \tag{109}$$

596   Define
$$v_{lj} = \frac{1}{n_l} \sum_{i=1}^{N} \mathbf{1}(\mathbf{X}_i \in B_l) \eta_j(\mathbf{X}_i), \tag{110}$$

597   then
$$\mathbb{E}[n_{lj} | \mathbf{X}_{1:N}] = n_l v_{lj}. \tag{111}$$

598   From Hoeffding's inequality,
$$\mathrm{P}(|n_{lj} - n_l v_{lj}| > t) \leq 2 e^{-\frac{1}{2n_l} t^2}. \tag{112}$$

599   Thus
$$\begin{aligned} \mathbb{E}\left[ \max_j |n_{lj} - n_l v_{lj}| \right] &= \int_0^\infty \mathrm{P}\left( \cup_{j=1}^{K} \{|n_{lj} - n_l v_{lj}| > t\} \right) dt \\ &\leq \int_0^\infty \min\left( 1, 2K e^{-\frac{1}{2n_l} t^2} \right) dt \\ &= \sqrt{2n_l \ln(2K)} + \int_{\sqrt{2n_l \ln(2K)}}^\infty 2K e^{-\frac{1}{2n_l} t^2} dt \\ &< 2\sqrt{2n_l \ln(2K)}, \end{aligned} \tag{113}$$

600   in which the last step uses the inequalit $\int_t^\infty e^{-u^2/(2\sigma^2)} du \leq \sqrt{2\pi}\sigma e^{-t^2/(2\sigma^2)}$. Then
$$\begin{aligned} \mathbb{E}[v_l^* - v_{lc_l} | \mathbf{X}_{1:N}] &= \frac{1}{n_l} \mathbb{E}[n_l v_l^* - n_l v_{lc_l}] \\ &= \frac{1}{n_l} \mathbb{E}\left[ n_l^* - n_{lc_l} + n_l v_l^* - n_l^* + n_{lc_l} - n_l v_{lc_l} \right] \\ &\leq \frac{1}{n_l} \mathbb{E}[n_l^* - n_{lc_l}] + \frac{2}{n_l} \mathbb{E}\left[ \max_j |n_{lj} - n_l v_{lj}| \right] \\ &\leq \frac{2}{\epsilon n_l}(\ln K + 1) + 4\sqrt{\frac{2\ln(2K)}{n_l}}. \end{aligned} \tag{114}$$

By Hölder continuity assumption (Assumption 1(a)), for $\mathbf{x} \in B_l$,

$$|v_{lj} - \eta_j(\mathbf{x})| \leq \frac{1}{n_l} \sum_{i=1}^{N} \mathbf{1}(\mathbf{X}_i \in B_l)|\eta_j(\mathbf{X}_i) - \eta_j(\mathbf{x})| \leq L_d h^\beta, \tag{115}$$

in which $L_d = L\sqrt{d}$, $L$ is the constant in Assumption 1(a). Thus

$$\mathbb{E}[\eta^*(\mathbf{x}) - \eta_{c_l}(\mathbf{x})|\mathbf{X}_{1:N}] \leq \frac{2}{\epsilon n_l}(\ln K + 1) + 4\sqrt{\frac{2\ln(2K)}{n_l}} + 2L_d h^\beta. \tag{116}$$

Now take integration over $B_l$.

$$\int_{B_l} (\eta^*(\mathbf{x}) - \mathbb{E}[\eta_{c_l}(\mathbf{x})]) f(\mathbf{x})d\mathbf{x}$$

$$\leq \ \mathrm{P}\left(n_l < \frac{1}{2}Np(B_l)\right)\int_{B_l}\left(\eta^*(\mathbf{x}) - \mathbb{E}\left[\eta_{c_l}(\mathbf{x})|n_l < \frac{1}{2}Np(B_l)\right]\right)f(\mathbf{x})d\mathbf{x}$$

$$+ \int_{B_l}\left(\eta^*(\mathbf{x}) - \mathbb{E}\left[\eta_{c_l}(\mathbf{x})|n_l \geq \frac{1}{2}Np(B_l)\right]\right)f(\mathbf{x})d\mathbf{x}$$

$$\leq \ p(B_l)\exp\left[-\frac{1}{2}(1 - \ln 2)Np(B_l)\right] + \left[\frac{2(\ln K + 1)}{\epsilon Np(B_l)} + 4\sqrt{\frac{2\ln(2K)}{Np(B_l)}} + 2L_d h^\beta\right]p(B_l), \tag{117}$$

in which $p(B_l) = \mathrm{P}(\mathbf{X} \in B_l) = \int_{B_l} f(\mathbf{x})d\mathbf{x}$. (117) is the central label DP counterpart of (67). The remainder of the proof follows arguments of the local label DP. We omit detailed steps. The result is

$$R - R^* \lesssim \left(\frac{\ln K}{\epsilon Nh^d} + \sqrt{\frac{\ln K}{Nh^d}} + h^\beta\right)^{\gamma+1}. \tag{118}$$

Let

$$h \sim \left(\frac{\ln K}{\epsilon N}\right)^{\frac{1}{\beta+d}} + \left(\frac{\ln K}{N}\right)^{\frac{1}{2\beta+d}}, \tag{119}$$

then

$$R - R^* \lesssim \left(\frac{\ln K}{\epsilon N}\right)^{\frac{\beta(\gamma+1)}{\beta+d}} + \left(\frac{\ln K}{N}\right)^{\frac{\beta(\gamma+1)}{2\beta+d}}. \tag{120}$$

The proof is complete.

## F   Proof of Theorem 5

Find $G$ cubes in the support and the length of each cube is $h$. Let $\phi(\mathbf{u})$ be the same as the classification case shown in appendix B. For $\mathbf{v} \in \mathcal{V} := \{-1, 1\}^G$, let

$$\eta_{\mathbf{v}}(\mathbf{x}) = \sum_{k=1}^{K} v_k \phi\left(\frac{\mathbf{x} - \mathbf{c}_k}{h}\right) h^\beta. \tag{121}$$

Let $\mathrm{P}(Y = 1|\mathbf{x}) = (1 + \eta_{\mathbf{v}}(\mathbf{x}))/2$, $\mathrm{P}(Y = -1|\mathbf{x}) = (1 - \eta_{\mathbf{v}}(\mathbf{x}))$, then $\eta(\mathbf{x}) = \mathbb{E}[Y|\mathbf{x}] = \eta_{\mathbf{v}}(\mathbf{x})$.

The overall volume of the support is bounded. Thus, we have

$$G \leq C_G h^{-d} \tag{122}$$

for some constant $C_G$.

Denote

$$\hat{v}_k = \mathrm{sign}\left(\int_{B_k} \hat{\eta}(\mathbf{x})\phi\left(\frac{\mathbf{x} - \mathbf{c}_k}{h}\right)f(\mathbf{x})d\mathbf{x}\right), \tag{123}$$

then the excess risk is bounded by

$$
\begin{aligned}
R &= \mathbb{E}\left[ (\hat{\eta}(\mathbf{X}) - \eta_{\mathbf{v}}(\mathbf{X}))^2 \right] \\
&= \sum_{k=1}^{K} \int_{B_k} \mathbb{E}\left[ (\hat{\eta}(\mathbf{x}) - \eta_{\mathbf{v}}(\mathbf{x}))^2 \right] f(\mathbf{x}) d\mathbf{x}.
\end{aligned}
\tag{124}
$$

If $\hat{v}_k \neq v_k$, from (123),

$$
\int_{B_k} \left( \hat{\eta}(\mathbf{x}) - v_k \phi \left( \frac{\mathbf{x} - \mathbf{c}_k}{h} \right) h^\beta \right)^2 f(\mathbf{x}) d\mathbf{x} \geq \int_{B_k} \left( \hat{\eta}(\mathbf{x}) + v_k \phi \left( \frac{\mathbf{x} - \mathbf{c}_k}{h} \right) h^\beta \right)^2 f(\mathbf{x}) d\mathbf{x}. \tag{125}
$$

Therefore, if $\hat{v}_k \neq v_k$, then

$$
\int_{B_k} (\hat{\eta}(\mathbf{x}) - \eta_{\mathbf{v}}(\mathbf{x}))^2 \, d\mathbf{x} \geq \frac{1}{2} \int_{B_k} \phi^2 \left( \frac{\mathbf{x} - \mathbf{c}_k}{h} \right) h^{2\beta} f(\mathbf{x}) d\mathbf{x} = \frac{1}{2} c h^{2\beta+d} \|\phi\|_2^2 . \tag{126}
$$

Therefore

$$
\begin{aligned}
R - R^* &\geq \mathbb{E}\left[ \frac{1}{2} c h^{2\beta+d} \|\phi\|_2^2 \mathbf{1}(\hat{v}_k \neq v_k) \right] \\
&= \frac{1}{2} c h^{2\beta+d} \|\phi\|_2^2 \, \mathbb{E}[\rho_H(\hat{\mathbf{v}}, \mathbf{v})].
\end{aligned}
\tag{127}
$$

Similar to the classification problem analyzed in Appendix B, let

$$
h \sim \left( N(\epsilon \wedge 1)^2 \right)^{-\frac{1}{2\beta+d}} , \tag{128}
$$

then $\delta \lesssim 1$, and

$$
\inf_{\hat{\mathbf{v}}} \sup_{M \in \mathcal{M}_\epsilon} \max_{\mathbf{v} \in \mathcal{V}} \mathbb{E}[\rho_H(\hat{\mathbf{v}}, \mathbf{v})] \gtrsim G \sim h^{-d}. \tag{129}
$$

Thus

$$
\inf_{\hat{\eta}} \inf_{M \in \mathcal{M}_\epsilon} \sup_{P_{X,Y} \in \mathcal{F}_{reg1}} R \gtrsim h^{2\eta+d} h^{-d} \sim h^{2\beta} \sim (N(\epsilon \wedge 1)^2)^{-\frac{2\beta}{2\beta+d}}. \tag{130}
$$

# G   Proof of Theorem 6

According to Assumption 2, $|Y| < T$ with probability 1, thus $\text{Var}[Y|\mathbf{x}] \leq T^2$ for any $\mathbf{x}$. A Laplacian distribution with parameter $\lambda$ has variance $2\lambda^2$, thus

$$
\text{Var}[W] = 2\lambda^2 = 2 \left( \frac{2T}{\epsilon} \right)^2 = \frac{8T^2}{\epsilon^2}. \tag{131}
$$

Hence

$$
\text{Var}[Z] = \text{Var}[Y] + \text{Var}[W] \leq T^2 \left( 1 + \frac{8}{\epsilon^2} \right). \tag{132}
$$

Now we analyze the bias first.

$$
\mathbb{E}[\hat{\eta}(\mathbf{x})] = \mathbb{E}\left[ \frac{1}{k} \sum_{i \in \mathcal{N}_k(\mathbf{x})} Z_i \right] = \mathbb{E}\left[ \frac{1}{k} \sum_{i \in \mathcal{N}_k(\mathbf{x})} \eta(\mathbf{X}_i) \right]. \tag{133}
$$

628  Thus

$$
\begin{aligned}
|\mathbb{E}[\hat{\eta}(\mathbf{x})] - \eta(\mathbf{x})| \;&\leq\; \mathbb{E}\left[\frac{1}{k}\sum_{i\in\mathcal{N}_k(\mathbf{x})}|\eta(\mathbf{X}_i)-\eta(\mathbf{x})|\right] \\[2mm]
&\leq\; \mathbb{E}\left[\frac{1}{k}\sum_{i\in\mathcal{N}_k(\mathbf{x})}\min\left\{L\|\mathbf{X}_i-\mathbf{x}\|^{\beta},2T\right\}\right] \\[2mm]
&\leq\; \mathbb{E}\left[\frac{1}{k}\sum_{i\in\mathcal{N}_k(\mathbf{x})}\min\left\{L\rho^{\beta}(\mathbf{x}),2T\right\}\right] \\[2mm]
&\leq\; 2T\mathrm{P}(\rho(\mathbf{x})>r_0)+Lr_0^{\beta} \\[2mm]
&\leq\; 2Te^{-(1-\ln 2)k}+L\left(\frac{2k}{Ncv_d\theta}\right)^{\frac{\beta}{d}} \\[2mm]
&\leq\; C_1\left(\frac{k}{N}\right)^{\frac{\beta}{d}},
\end{aligned}
\tag{134}
$$

629  for some constant $C_1$.

630  It remains to bound the variance.

$$
\mathrm{Var}[\hat{\eta}(\mathbf{x})]=\mathbb{E}\left[\mathrm{Var}\left[\hat{\eta}(\mathbf{x})|\mathbf{X}_1,\ldots,\mathbf{X}_N\right]\right]+\mathrm{Var}[\mathbb{E}[\hat{\eta}(\mathbf{x})]|\mathbf{X}_1,\ldots,\mathbf{X}_N].
\tag{135}
$$

631  For the first term in (135),

$$
\begin{aligned}
\mathrm{Var}[\hat{\eta}(\mathbf{x})|\mathbf{X}_1,\ldots,\mathbf{X}_N] \;&=\; \mathrm{Var}\left[\frac{1}{k}\sum_{i\in\mathcal{N}_k(\mathbf{x})}Z_i|\mathbf{X}_1,\ldots,\mathbf{X}_N\right] \\[2mm]
&=\; \frac{1}{k^2}\sum_{i\in\mathcal{N}_k(\mathbf{x})}\mathrm{Var}[Z_i|\mathbf{X}_1,\ldots,\mathbf{X}_N] \\[2mm]
&\leq\; \frac{1}{k}T^2\left(1+\frac{8}{\epsilon^2}\right).
\end{aligned}
\tag{136}
$$

632  For the second term in (135),

$$
\begin{aligned}
\mathrm{Var}[\mathbb{E}[\hat{\eta}(\mathbf{x})|\mathbf{X}_1,\ldots,\mathbf{X}_N]] \;&=\; \mathrm{Var}\left[\frac{1}{k}\sum_{i\in\mathcal{N}_k(\mathbf{x})}\eta(\mathbf{X}_i)\right] \\[2mm]
&\leq\; \mathbb{E}\left[\left(\frac{1}{k}\sum_{i\in\mathcal{N}_k(\mathbf{x})}\eta(\mathbf{X}_i)-\eta(\mathbf{x})\right)^2\right] \\[2mm]
&=\; \frac{1}{k}\sum_{i\in\mathcal{N}_k(\mathbf{x})}\mathbb{E}\left[(\eta(\mathbf{X}_i)-\eta(\mathbf{x}))^2\right] \\[2mm]
&\leq\; \frac{1}{k}\sum_{i\in\mathcal{N}_k(\mathbf{x})}\mathbb{E}\left[\min\left\{L^2\|\mathbf{X}_i-\mathbf{x}\|^{2\beta},4T^2\right\}\right] \\[2mm]
&\leq\; 4T^2e^{-(1-\ln 2)k}+L^2r_0^{2\beta} \\[2mm]
&\leq\; C_1^2\left(\frac{k}{N}\right)^{\frac{2\beta}{d}}.
\end{aligned}
\tag{137}
$$

633  Therefore (135) becomes

$$
\mathrm{Var}[\hat{\eta}(\mathbf{x})]\leq\frac{1}{k}T^2\left(1+\frac{8}{\epsilon^2}\right)+C_1^2\left(\frac{k}{N}\right)^{\frac{2\beta}{d}}.
\tag{138}
$$

634 Combine the analysis of bias and variance,

$$\mathbb{E}[(\hat{\eta}(\mathbf{x}) - \eta(\mathbf{x}))^2] \leq \frac{1}{k}T^2\left(1 + \frac{8}{\epsilon^2}\right) + 2C_1^2\left(\frac{k}{N}\right)^{\frac{2\beta}{d}}. \tag{139}$$

635 Therefore the overall risk is bounded by

$$R = \mathbb{E}[(\hat{\eta}(\mathbf{X}) - \eta(\mathbf{X}))^2] \lesssim \frac{1}{k}T^2\left(1 + \frac{8}{\epsilon^2}\right) + 2C_1^2\left(\frac{k}{N}\right)^{\frac{2\beta}{d}}. \tag{140}$$

636 The optimal growth rate of $k$ over $N$ is

$$k \sim N^{\frac{2\beta}{d+2\beta}}(\epsilon \wedge 1)^{-\frac{2d}{d+2\beta}}. \tag{141}$$

637 Then the convergence rate of the overall risk becomes

$$R \lesssim (N(\epsilon \wedge 1)^2)^{-\frac{2\beta}{d+2\beta}}. \tag{142}$$

# H    Proof of Theorem 7

639 From (127),

$$\begin{aligned}
R - R^* &\geq \frac{1}{2}ch^{2\beta+d}\|\phi\|_2^2\,\mathbb{E}[\rho_H(\hat{\mathbf{V}}, \mathbf{V})] \\
&= \frac{1}{2}ch^{2\beta+d}\|\phi\|_2^2\sum_{k=1}^{G}\mathrm{P}(\hat{V}_k \neq V_k).
\end{aligned} \tag{143}$$

640 Follow the analysis of lower bounds of classification in Appendix D, let $h$ scales as (101), then
641 $\mathrm{P}(\hat{V}_k \neq V_k) \gtrsim 1$. Moreover, $G \sim h^{-d}$. Hence

$$\inf_{\mathcal{A}\in\mathcal{A}_\epsilon}\sup_{(f,\eta)\in\mathcal{F}_{reg1}}(R - R^*) \gtrsim h^{2\beta} \sim (\epsilon N)^{-\frac{2\beta}{d+\beta}}. \tag{144}$$

642 Moreover, note that the non-private lower bound of regression is

$$\inf_{\mathcal{A}\in\mathcal{A}_\epsilon}\sup_{(f,\eta)\in\mathcal{F}_{reg1}}(R - R^*) \gtrsim N^{-\frac{2\beta}{2\beta+d}}. \tag{145}$$

643 Combine (144) and (145),

$$\inf_{\mathcal{A}\in\mathcal{A}_\epsilon}\sup_{(f,\eta)\in\mathcal{F}_{reg1}}(R - R^*) \gtrsim N^{-\frac{2\beta}{2\beta+d}} + (\epsilon N)^{-\frac{2\beta}{d+\beta}}. \tag{146}$$

# I    Proof of Theorem 8

645 *1) Analysis of bias.* Note that

$$\mathbb{E}[\hat{\eta}_l|\mathbf{X}_{1:N}] = \mathbb{E}[Y|\mathbf{X} \in B_l] = \frac{1}{p(B_l)}\int\eta(\mathbf{u})f(\mathbf{u})d\mathbf{u}. \tag{147}$$

646 Therefore, for all $\mathbf{x} \in B_l$,

$$\begin{aligned}
|\mathbb{E}[\hat{\eta}_l|\mathbf{X}_{1:N}] - \eta(\mathbf{x})| &\leq \frac{1}{p(B_l)}\int|\eta(\mathbf{u}) - \eta(\mathbf{x})|f(\mathbf{u})d\mathbf{u} \\
&\leq L_d h^\beta.
\end{aligned} \tag{148}$$

647 Therefore for all $\mathbf{x} \in B_l$,

$$|\mathbb{E}[\hat{\eta}_l] - \eta(\mathbf{x})| \leq L_d h^\beta. \tag{149}$$

648 *2) Analysis of variance.* If $n_l > 0$,

$$\mathrm{Var}\left[\frac{1}{n_l}\sum_{i=1}^{N}\mathbf{1}(\mathbf{X}_i \in B_l)Y_i\Big|\mathbf{X}_{1:N}\right] = \frac{1}{n_l}\mathrm{Var}[Y|\mathbf{X} \in B_l] \leq \frac{1}{n_l}. \tag{150}$$

Therefore

$$\mathrm{Var}\left[\frac{1}{n_l}\sum_{i=1}^{N}\mathbf{1}(\mathbf{X}_i\in B_l)Y_i\right] \leq \mathrm{P}\left(n_l<\frac{1}{2}Np(B_l)\right)+\mathrm{P}\left(n_l\geq\frac{1}{2}Np(B_l)\right)\frac{2}{Np(B_l)}$$
$$\leq \exp\left[-\frac{1}{2}(1-\ln 2)Np(B_l)\right]+\frac{2}{Nch^d}. \tag{151}$$

Similarly,

$$\mathrm{Var}[W_l] \leq \mathrm{P}\left(n_l<\frac{1}{2}Np(B_l)\right)\frac{1}{\epsilon^2}+\mathrm{P}\left(n_l\geq\frac{1}{2}Np(B_l)\right)\frac{8}{\left(\frac{1}{2}Np(B_l)\right)^2\epsilon^2}$$
$$\lesssim \frac{1}{N^2h^{2d}\epsilon^2}. \tag{152}$$

The mean squared error can then be bounded by the bounds of bias and variance.

$$\mathbb{E}\left[(\hat{\eta}(\mathbf{x})-\eta(\mathbf{x}))^2\right]\lesssim h^{2\beta}+\frac{1}{Nh^d}+\frac{1}{N^2h^{2d}\epsilon^2}. \tag{153}$$

Let

$$h\sim N^{-\frac{1}{2\beta+d}}+(\epsilon N)^{-\frac{1}{d+\beta}}. \tag{154}$$

Then

$$R-R^*\lesssim N^{-\frac{2\beta}{2\beta+d}}+(\epsilon N)^{-\frac{2\beta}{d+\beta}}. \tag{155}$$

## J  Proof of Theorem 9

Now we prove the minimax lower bound of nonparametric regression under label DP constraint. We focus on the case in which $\epsilon$ is small.

Similar to the steps of the proof of Theorem 5 in Appendix F, we find $B$ cubes in the support. The definition of $\eta_{\mathbf{v}}$, $\hat{v}_k$ are also the same as (121) and (123). Compared with the case with bounded noise, now $Y$ can take values in $\mathbb{R}$.

For given $\mathbf{x}$, let

$$Y=\begin{cases} T & \text{with probability} & \frac{1}{2}\left(\frac{M_p}{T^p}+\frac{\eta_{\mathbf{v}}(\mathbf{x})}{T}\right) \\ 0 & \text{with probability} & 1-\frac{M_p}{T^p} \\ -T & \text{with probability} & \frac{1}{2}\left(\frac{M_p}{T^p}-\frac{\eta_{\mathbf{v}}(\mathbf{x})}{T}\right). \end{cases} \tag{156}$$

In Appendix F about the case with bounded noise, $T$ is a fixed constant. However, here $T$ is not fixed and will change over $N$. It is straightforward to show that the distribution of $Y$ in (156) satisfies Assumption 3:

$$\mathbb{E}[|Y|^p|\mathbf{x}]=M_p. \tag{157}$$

Moreover, by taking expectation over $Y$, it can be shown that $\eta_{\mathbf{v}}$ is still the regression function:

$$\mathbb{E}[Y|\mathbf{x}]=\eta_{\mathbf{v}}(\mathbf{x}). \tag{158}$$

Let

$$T=\left(\frac{1}{2}M_ph^{-\beta}\right)^{\frac{1}{p-1}}. \tag{159}$$

Here we still define

$$\delta=\sup_{M\in\mathcal{M}_\epsilon}\max_{\mathbf{v},\mathbf{v}':\rho_H(\mathbf{v},\mathbf{v}')=1}D(P_{(X,Z)_{1:N}|\mathbf{v}}||P_{(X,Z)_{1:N}|\mathbf{v}'}). \tag{160}$$

667    Without loss of generality, suppose that $v_1 = v_1'$ for $i \neq 1$. Then

$$
\begin{aligned}
D(P_{(X,Z)_{1:N}|\mathbf{v}} || P_{(X,Z)_{1:N}|\mathbf{v}'}) &= ND(P_{X,Z|\mathbf{v}} || P_{X,Z|\mathbf{v}'}) \\
&= N \int_{B_1} f(\mathbf{x}) D(P_{Z|\mathbf{X},\mathbf{v}} || P_{Z|\mathbf{X},\mathbf{v}'}) d\mathbf{x} \\
&\leq N \int_{B_1} f(\mathbf{x}) (e^\epsilon - 1)^2 \mathbb{TV}^2 \left( P_{Z|X,\mathbf{v}}, P_{Z|X,\mathbf{v}'} \right) d\mathbf{x} \\
&= N \int_{B_1} f(\mathbf{x}) (e^\epsilon - 1)^2 \eta_{\mathbf{v}}^2(\mathbf{x}) \frac{1}{T^2} d\mathbf{x} \\
&= N (e^\epsilon - 1)^2 \frac{h^{2\beta}}{T^2} \int_{B_1} f(\mathbf{x}) \phi^2 \left( \frac{\mathbf{x} - \mathbf{c}_1}{h} \right) d\mathbf{x} \\
&= N (e^\epsilon - 1)^2 h^{2\beta + d} \|\phi\|_2^2 T^{-2} \\
&= N (e^\epsilon - 1)^2 \|\phi\|_2^2 \left( \frac{1}{2} M_p \right)^{-\frac{2}{p-1}} h^{2\beta + d + \frac{2\beta}{p-1}}.
\end{aligned} \tag{161}
$$

668    Let

$$
h \sim (N(e^\epsilon - 1)^2)^{-\frac{p-1}{2p\beta + d(p-1)}}, \tag{162}
$$

669    then $\delta \lesssim 1$. Hence

$$
\inf_{\hat{\eta}} \inf_{M \in \mathcal{M}_\epsilon} \sup_{(f,\eta) \in \mathcal{F}} R \gtrsim h^{2\beta} \sim (N(e^\epsilon - 1)^2)^{-\frac{2\beta(p-1)}{2p\beta + d(p-1)}}. \tag{163}
$$

## K   Proof of Theorem 10

671    Define

$$
\eta_T(\mathbf{x}) := \mathbb{E}[Y_T | \mathbf{x}]. \tag{164}
$$

672    Then

$$
\hat{\eta}(\mathbf{x}) - \eta(\mathbf{x}) = \eta_T(\mathbf{x}) - \eta(\mathbf{x}) + \mathbb{E}[\hat{\eta}(\mathbf{x})] - \eta_T(\mathbf{x}) + \hat{\eta}(\mathbf{x}) - \mathbb{E}[\hat{\eta}(\mathbf{x})]. \tag{165}
$$

673    Therefore

$$
\begin{aligned}
\mathbb{E}\left[ (\hat{\eta}(\mathbf{x}) - \eta(\mathbf{x}))^2 \right] &\leq 3(\eta_T(\mathbf{x}) - \eta(\mathbf{x}))^2 + 3(\mathbb{E}[\hat{\eta}(\mathbf{x})] - \eta_T(\mathbf{x}))^2 + 3 \operatorname{Var}[\hat{\eta}(\mathbf{x})] \\
&:= 3(I_1 + I_2 + I_3).
\end{aligned} \tag{166}
$$

674    Now we bound $I_1$, $I_2$ and $I_3$ separately.

675    **Bound of $I_1$.** We show the following lemma (which will also be used later).

**Lemma 2.**

$$
|\eta_T(\mathbf{x}) - \eta(\mathbf{x})| \leq \frac{M_p}{p-1} T^{1-p}. \tag{167}
$$

676    *Proof.* Firstly, we decompose $\eta_T(\mathbf{x})$ and $\eta(\mathbf{x})$:

$$
\eta_T(\mathbf{x}) = \mathbb{E}[Y_T | \mathbf{x}] = \mathbb{E}[Y \mathbf{1}(-T \leq Y \leq T) | \mathbf{x}] + T P(Y > T | \mathbf{x}) - T P(Y < T | \mathbf{x}), \tag{168}
$$

677

$$
\eta(\mathbf{x}) = \mathbb{E}[Y | \mathbf{x}] = \mathbb{E}[Y \mathbf{1}(-T \leq Y \leq T) | \mathbf{x}] + \mathbb{E}[Y \mathbf{1}(Y > T) | \mathbf{x}] - \mathbb{E}[Y \mathbf{1}(Y < T) | \mathbf{x}]. \tag{169}
$$

678    The first term is the same between (168) and (169). Therefore we only need to compare the second
679    and the third term.

$$
\begin{aligned}
\mathbb{E}[Y \mathbf{1}(Y > T) | \mathbf{x}] &= \int_0^T P(Y > T | \mathbf{x}) dt + \int_T^\infty P(Y > T | \mathbf{x}) dt \\
&\leq T P(Y > T | \mathbf{x}) + \int_T^\infty M_p t^{-p} dt \\
&= T P(Y > T | \mathbf{x}) + \frac{M_p}{p-1} T^{1-p}.
\end{aligned} \tag{170}
$$

Therefore

$$\mathbb{E}[Y\mathbf{1}(Y > T)|\mathbf{x}] - T\mathrm{P}(Y > T|\mathbf{x}) \le \frac{M_p}{p-1}T^{1-p}. \tag{171}$$

Similarly,

$$TP(Y < T|\mathbf{x}) - \mathbb{E}[Y\mathbf{1}(Y < T)|\mathbf{x}] \le \frac{M_p}{p-1}T^{1-p}. \tag{172}$$

A Combination of these two inequalities yields the (167). □

With Lemma 2,

$$I_1 \le \frac{M_p^2}{(p-1)^2}T^{2(1-p)}. \tag{173}$$

**Bound of $I_2$.** Follow the steps in (134),

$$I_2 \le C_1^2 \left(\frac{k}{N}\right)^{\frac{2\beta}{d}}. \tag{174}$$

**Bound of $I_3$.** We decompose $\mathrm{Var}[\hat{\eta}(\mathbf{x})]$ as following:

$$\mathrm{Var}[\hat{\eta}(\mathbf{x})] = \mathbb{E}[\mathrm{Var}[\hat{\eta}(\mathbf{x})|\mathbf{X}_1,\ldots,\mathbf{X}_N]] + \mathrm{Var}[\mathbb{E}[\hat{\eta}(\mathbf{x})|\mathbf{X}_1,\ldots,\mathbf{X}_N]]. \tag{175}$$

For the first term in (175), from Assumption 3, $\mathbb{E}[|Y|^p|\mathbf{x}] \le M_p$. Since $p \ge 2$, we have $\mathbb{E}[Y^2|\mathbf{x}] = M_p^{\frac{2}{p}}$. Therefore

$$\mathrm{Var}[Z_i|\mathbf{X}_1,\ldots,\mathbf{X}_N] = \mathrm{Var}[Y_T] + \mathrm{Var}[W] \le M_p^{\frac{2}{p}} + \frac{8T^2}{\epsilon^2}. \tag{176}$$

Recall (20), we have

$$\begin{aligned}
\mathrm{Var}[\hat{\eta}(\mathbf{x})|\mathbf{X}_1,\ldots,\mathbf{X}_N] &= \frac{1}{k^2}\sum_{i\in\mathcal{N}_k(\mathbf{x})}\mathrm{Var}[Z_i|\mathbf{X}_1,\ldots,\mathbf{X}_N] \\
&\le \frac{1}{k}\left(M_p^{\frac{2}{p}} + \frac{8T^2}{\epsilon^2}\right).
\end{aligned} \tag{177}$$

For the second term in (175), (137) still holds, thus

$$\mathrm{Var}[\mathbb{E}[\hat{\eta}(\mathbf{x})|\mathbf{X}_1,\ldots,\mathbf{X}_N]] \le C_1^2\left(\frac{k}{N}\right)^{\frac{2\beta}{d}}, \tag{178}$$

and

$$I_3 \le \frac{1}{k}\left(M_p^{\frac{2}{p}} + \frac{8T^2}{\epsilon^2}\right) + C_1^2\left(\frac{k}{N}\right)^{\frac{2\beta}{d}}. \tag{179}$$

Plug (173), (174) and (179) into (166), and take expectations, we get

$$\begin{aligned}
R &= \mathbb{E}[(\hat{\eta}(\mathbf{X}) - \eta(\mathbf{X}))^2] \\
&\lesssim T^{2(1-p)} + \frac{1}{k} + \frac{T^2}{k\epsilon^2} + \left(\frac{k}{N}\right)^{\frac{2\beta}{d}}.
\end{aligned} \tag{180}$$

Let

$$T \sim (k\epsilon^2)^{\frac{1}{2p}}, k \sim (N\epsilon^2)^{\frac{2p\beta}{d(p-1)+2p\beta}} \vee N^{\frac{2\beta}{2\beta+d}}, \tag{181}$$

then

$$R \lesssim (N\epsilon^2)^{-\frac{2\beta(p-1)}{d(p-1)+2p\beta}} \vee N^{-\frac{2\beta}{2\beta+d}}. \tag{182}$$

 # L Proof of Theorem 11

Let $Y$ be distributed as (156). Recall Lemma 1 for the problem of classification and regression with bounded noise.

Now we show the corresponding lemma for regression with unbounded noise.

**Lemma 3.** *If $0 \le t \le T \ln 2/(\epsilon n_k)$, and $n_k t/T$ is an integer, then*

$$P(\hat{V}_k = 1|\mathbf{X}_{1:N}, \bar{Y}_k = -t) + P(\hat{V}_k = -1|\mathbf{X}_{1:N}, \bar{Y}_k = t) \ge \frac{2}{3}. \tag{183}$$

Here we briefly explain the condition $n_k t$ is an integer. Recall the definition of $\bar{Y}_k$ in (88). Now since $Y$ take values in $\{-T, 0, T\}$, $n_k \bar{Y}_k/T$ must be an integer. Therefore, in Lemma 3, we only need to consider the case such that $n_k t/T$ is an integer.

*Proof.* The proof follows the proof of Lemma 1 closely. We provide the proof here for completeness.

Construct $D'$ by changing the label values of $l = n_k t/T$ items from these $n_k$ samples falling in $B_k$, from $-T$ to $T$. Then the average label values in $B_k$ is denoted as $\bar{Y}_k'$ after such replacement. $\hat{V}_k$ also becomes $\hat{V}_k'$. Then from the $\epsilon$-label DP requirement,

$$
\begin{aligned}
P(\hat{V}_k = 1|\mathbf{X}_{1:N}, \bar{Y}_k = -t) &\overset{(a)}{\ge} e^{-l\epsilon} P\left(\hat{V}_k' = 1|\mathbf{X}_{1:N}, \bar{Y}_k' = -t + \frac{2l}{n_k}\right) \\
&\overset{(b)}{\ge} e^{-l\epsilon} P\left(\hat{V}_k = 1|\mathbf{X}_{1:N}, \bar{Y}_k = -t + \frac{2l}{n_k}\right) \\
&\ge e^{-n_k t\epsilon}\left[1 - P\left(\hat{V}_k = -1|\mathbf{X}_{1:N}, \bar{Y}_k = -t + \frac{2l}{n_k}\right)\right] \\
&\ge \frac{1}{2}\left[1 - P\left(\hat{V}_k = -1|\mathbf{X}_{1:N}, \bar{Y}_k = t\right)\right]. \tag{184}
\end{aligned}
$$

in which (a) uses the group privacy property. The Hamming distance between $D$ and $D'$ is $l$, thus the ratio of probability between $D$ and $D'$ is within $[e^{-l\epsilon}, e^{l\epsilon}]$. (b) holds because the algorithm does not change after changing $D$ to $D'$. Similarly,

$$P(\hat{V}_k = -1|\mathbf{X}_{1:N}, \bar{Y}_k = t) \ge \frac{1}{2}\left[1 - P\left(\hat{V}_k = 1|\mathbf{X}_{1:N}, \bar{Y}_k = -t\right)\right]. \tag{185}$$

Then (183) can be shown by adding up (184) and (185). □

We then follow the proof of Theorem 3 in Appendix D. (101) becomes

$$h \sim \left(\frac{\epsilon N}{T}\right)^{-\frac{1}{d+\beta}}. \tag{186}$$

In (156), note that $P(Y = T) \ge 0$ and $P(Y = -T) \ge 0$. Therefore $M_p/T^p \ge \eta_{\mathbf{v}}(\mathbf{x})/T$. This requires $h^\beta T^{p-1} \le M_p$. Let $T \sim h^{-\frac{\beta}{p-1}}$, then

$$h \sim (\epsilon N)^{-\frac{1}{d+\beta}} h^{\frac{\beta}{(d+\beta)(p-1)}}, \tag{187}$$

i.e.

$$h \sim (\epsilon N)^{-\frac{p-1}{p\beta+d(p-1)}}. \tag{188}$$

Combine with standard minimax rate, the lower bound of regression with unbounded noise is

$$\inf_{\mathcal{A} \in \mathcal{A}_\epsilon} \sup_{(f,\eta) \in \mathcal{F}_{reg2}} (R - R^*) \gtrsim N^{-\frac{2\beta}{2\beta+d}} + (\epsilon N)^{-\frac{2\beta(p-1)}{p\beta+d(p-1)}}. \tag{189}$$

## M  Proof of Theorem 12

*1) Analysis of bias.* Note that Lemma 2 still holds here. Moreover, recall (149). Therefore

$$|\mathbb{E}[\hat{\eta}_l] - \eta(\mathbf{x})| \leq |\mathbb{E}[\hat{\eta}_l - \eta_T(\mathbf{x})]| + |\eta_T(\mathbf{x}) - \eta(\mathbf{x})| \leq L_d h^\beta + \frac{M_p}{p-1} T^{1-p}. \tag{190}$$

*2) Analysis of variance.* Similar to (151), it can be shown that

$$\mathrm{Var}\left[\frac{1}{n_l} \sum_{i=1}^N \mathbf{1}(\mathbf{X}_i \in B_l) Y_i\right] \lesssim \frac{1}{N h^d}. \tag{191}$$

Moreover, the noise variance can be bounded by

$$\mathrm{Var}[W_l] \lesssim \frac{T^2}{N^2 h^{2d} \epsilon^2}. \tag{192}$$

The mean squared error is then bounded by

$$\mathbb{E}\left[(\hat{\eta}(\mathbf{x}) - \eta(\mathbf{x}))^2\right] \lesssim h^{2\beta} + T^{2(1-p)} + \frac{T^2}{N^2 h^{2d} \epsilon^2} + \frac{1}{N h^d}. \tag{193}$$

Let $T \sim (\epsilon N h^d)^{1/p}$, then

$$R - R^* = \mathbb{E}\left[(\hat{\eta}(\mathbf{X}) - \eta(\mathbf{X}))^2\right] \lesssim h^{2\beta} + \frac{1}{N h^d} + (\epsilon N h^d)^{-2(1-1/p)}. \tag{194}$$

To minimize (194), let

$$h \sim N^{-\frac{1}{2\beta+d}} + (\epsilon N)^{-\frac{p-1}{p\beta+d(p-1)}}, \tag{195}$$

then

$$R - R^* \lesssim N^{-\frac{2\beta}{2\beta+d}} + (\epsilon N)^{-\frac{2\beta(p-1)}{p\beta+d(p-1)}}. \tag{196}$$

