# OpenReview forum: "On Theoretical Limits of Learning with Label Differential Privacy"
_NeurIPS.cc/2024/Conference — Submitted to NeurIPS 2024_

### Official Review · Reviewer_WgYH · 2024-07-03

**Soundness:** 3
**Presentation:** 2
**Contribution:** 3
**Rating:** 6
**Confidence:** 3

**Summary:**

This paper investigates the theoretical boundaries of learning with Label Differential Privacy (Label-DP) in both central and local models.
Label-DP is a weakening of standard differential privacy, where only the privacy of the "label" of each example is to be protected (an example is a pair (feature vector, label)).

The key contributions of the paper are to establish min-max optimal rates for excess error in the settings of:
* (multi-class) classification,
* regression with bounded labels,
* regression with unbounded labels (but under a bounded moment condition).

The min-max rates are over the class of data distributions that satisfy $\beta$-Holder smoothness, admits a lower bound on probability density that is bounded away from zero, assumes that there are no “sharp corners” in the input space, and a $\gamma$-margin assumption (in case of classification), or bounded label range or bounded label moments (in case of regression).

These min-max rates are then compared against the previously known min-max rates for learning under “full” local-DP (that protects both features and labels), as well as non-private learning.

The key takeaways are:
* Local-DP vs Non-Private:
  * For classification and regression with bounded labels, the sample complexity under Local-DP increases by a factor of $1/\varepsilon^2$, but has the same rate in terms of desired excess error. This is unlike “Full Local-DP”, where the sample complexity is larger even in terms of the desired excess error.
  * For regression with unbounded labels, the dependence of sample complexity on desired excess error is worse than the non-private setting.
* Central-DP vs Non-Private:
  * The excess error is the sum of the non-private excess error and an additional term that decays faster in the number of samples, so the additional sample complexity due to privacy is negligible for very small excess error.

**Strengths:**

The paper provides a comprehensive study of the min-max rates for learning under label differential privacy, in both local and central models of DP, and for both classification and regression. This complements prior literature on min-max rates for learning (non-privately) and for learning under (full) differential privacy. The rates highlight the precise cost of _label_ differential privacy and the sample complexity benefits over full differential privacy.

**Weaknesses:**

While there are many results in the paper, I think the proof techniques in both lower and upper bounds use mostly standard tools (This is not necessarily a weakness!).

The paper writing could be improved at several places though. Some comments are listed below under "Questions".

**Questions:**

I don't have any significant questions.

Some suggestions about writing are listed below:
* Table 1: I understood that the last two rows of Table 1 are prior work, and the first two rows are the new contributions in this paper. But it would be good to include appropriate citations prominently, e.g. in the caption for Table 1.
* Line 122: $\mathbb{E}[l(g(X, Y))]$ should be $\mathbb{E}[l(g(X), Y)]$.
* Line 135 (Eq 8): I think $\eta^*(x) = \max_k \eta_k(x)$, but it is not formally defined anywhere?
* Line 137: Proof of Proposition 2 in Appendix A does not cover the regression setting. (It is quite standard, but might be good to include the proof for completeness.)
* Line 139: Is Assumption 1 only about the classification setting? If so, maybe it makes sense to move it to Section 4?
* Line 145: Assumption 1 (d): It is not explained until this point that $\mathcal{X} \subseteq \mathbb{R}^d$. I think this should be defined upfront if this is assumed throughout the paper.
* Line 169: If I understand correctly, the $\inf_{\hat{Y}}$ refers to $\hat{Y}$ that is a function of the outputs of the mechanism on all inputs $(M(x_i, y_i))_i$. This notation is implicit, but would be good to spell out.
* Line 202: Is there a reference for the bounds for “full” local DP, that protects both features and labels ?
* Line 244: How is Assumption 1 relevant for regression? What is $\eta_j$ ? Only points (c) and (d) make sense. Maybe (a) also makes sense for $\eta$ instead of $\eta_j$ ?
* Line 297: $Y_{Ti}$ notation is awkward to read. Might want to make it something like $Y^T_i$?

**Limitations:**

I do not see any potential negative societal impact of this work, as it is primarily theoretical.

---

> ### Author Rebuttal · Authors · 2024-08-01
>
> Thank you very much for your careful reading and valuable suggestions.
>
> **Table 1**
>
> Thanks for this comment, which is also raised by another reviewer kpNG. We will add citations directly in the table.
>
> **Line 122**
>
> Thanks. We will correct this typo.
>
> **Line 135**
>
> Thanks. $\eta^*(x)=\max_k \eta_k(x)$ is the maximum value of regression functions.
>
> **Line 137**
>
> Thanks. Although the proof is straightforward, we will add the proof for completeness.
>
> **Line 139**
>
> Assumption 1(b) is for the classification setting only. (a), (c), (d) are also necessary for regression. We will revise the paper accordingly. Further discussions are provided in the reply of line 244.
>
> **Line 145**
>
> We will add the point that $\mathcal{X}\subset \mathbb{R}^d$.
>
> **Line 169**
>
> Yes, $\hat{Y}$ is a function of outputs of the mechanism on all inputs. This is not a typo, but we agree that it is better to emphasize it in the paper.
>
> **Line 202**
>
> This is the same as the comment about Table 1. Line 79-85 provide a brief overview of related works about full DP. The following are three main references about full DP.
>
> [1] Density estimation: Duchi, John C., Michael I. Jordan, and Martin J. Wainwright. "Minimax optimal procedures for locally private estimation." Journal of the American Statistical Association 2018.
>
> [2] Classification:  Berrett, T., C. Butucea. Classification under local differential privacy. arXiv:1912.04629, 2019.
>
> [3] Regression:  Berrett, T. B., L. Györfi, H. Walk. Strongly universally consistent nonparametric regression and classification with privatised data. Electronic Journal of Statistics, 15:2430–2453, 2021.
>
> The main technical tools are developed in [1]. [2] and [3] generalize the results to classification and regression problems.
>
> **Line 244**
>
> Here we continue the discussions about line 139. Yes, for assumption (a), here it is about $\eta$ instead of $\eta_j$: $|\eta(x)-\eta(x')|\leq L||x-x'||^\beta$.
>
> (c) and (d) remain the same for the regression case.
>
> We will revise the paper accordingly.
>
> **Line 297**
>
> Thanks for this suggestion. $Y_i^T$ appears to be better. We will change the notation in our revised paper.
>
> We thank again for your careful reading and fruitful suggestions. We will address these problems in our revised version.

---

> > ### Comment · Reviewer_WgYH · 2024-08-11
> > **Response to Rebuttal**
> >
> > I would like to thank the author(s) for their response. The proposed changes sound good to me, and I will keep my rating.

---

> > > ### Author Response · Authors · 2024-08-11
> > >
> > > Thank you very much again for your feedback and for engaging in this discussion. If you have any remaining questions, please do let us know.

---

### Official Review · Reviewer_vnjU · 2024-07-15

**Soundness:** 3
**Presentation:** 2
**Contribution:** 3
**Rating:** 7
**Confidence:** 2

**Summary:**

This work studies the minimax rates for classification and regression under (pure) label differential privacy in both the local and central models. They prove that rates of convergence for classification and regression with bounded label noise in the local label DP model are comparable to those for the non-private tasks, except for the expected $1/\varepsilon^2$ dependence. This represents an improvement over rates for standard DP in both settings, where there is a worse dependence on the dimension of the covariates. They also prove, however, in the case of regression with unbounded label noise, the convergence rate improvements over “full” DP aren’t as meaningful.

**Strengths:**

This work makes notable progress in our theoretical understanding of the costs of label DP relative to non-private and full DP algorithms for the same learning task.

**Weaknesses:**

The presentation could be improved in several places. Admittedly, this is written from a statistical perspective that is different from the one I am most familiar with, so some of the perceived presentation issues may just be a matter of convention, but the following changes might make this work more understandable to the general NeurIPS community:


Abstract:

The main challenge and the techniques to overcome them as stated in the abstract aren’t clear to me as a reader at this point. It’s not yet stated that the subject of interest is minimax rates, and so there’s no context for the statement “take infimum over all possible learners” and why that would present a challenge. Generally, I did not have a good idea of what the contribution of this work was from the abstract.

Introduction:

“the learning performances” -> “the learning performance”

“the label DP” -> “label DP”

In Table 1, attribution for the full DP rates in the local DP setting as well as the rates in the non-private setting should be given in the table. Also, I think there’s an issue with the parentheses in the local label DP rates for regression with bounded label noise.

Section 2:

In the “Minimax analysis for private data” paragraph, KNLRS11 is attributed with finding the relation between label DP and stochastic queries. This is not accurate, this work characterizes local DP learning by the statistical query model.

Section 3:

“We hope that $R - R^*$ to be as small as possible” -> “we seek to minimize this risk” or something similar

“the Bayes optimal classifier and the corresponding Bayes risk is” -> “the Bayes optimal classifier and the corresponding Bayes risk are”

In Proposition 2, f(x) is used before it is defined.

Section 4:

I didn’t find the proof outline for Theorem 1 or Theorem 3 to be informative at all. It would be good to add more specifics if possible.

“Let the privacy mechanism M(x,y) outputs” -> “Let the privacy mechanism M(x,y) output”

**Questions:**

Do you have a sense of how the rates would change for approximate label DP?

**Limitations:**

Yes, the authors address the limitations.

---

> ### Author Rebuttal · Authors · 2024-08-01
>
> Thank you very much for your positive review and detailed comments.
>
> Firstly, thanks for finding these grammatical errors. We will correct them during revision.
>
> **The main challenge and the techniques to overcome them as stated in the abstract aren’t clear to me as a reader at this point. It’s not yet stated that the subject of interest is minimax rates, and so there’s no context for the statement “take infimum over all possible learners” and why that would present a challenge. Generally, I did not have a good idea of what the contribution of this work was from the abstract.**
>
> "Take infimum over all possible learners" means that, standard packing method can only be used for statistical problems whose output dimension is fixed. However, to derive the minimax optimal risk of classification and regression, we need to consider all possible learners with arbitrary dimensionality. We will improve the abstract accordingly.
>
> **In Table 1, attribution for the full DP rates in the local DP setting as well as the rates in the non-private setting should be given in the table. Also, I think there’s an issue with the parentheses in the local label DP rates for regression with bounded label noise.**
>
> Thanks for this comment. Other reviewers also mention adding references to the table. Moreover, the second right parenthesis need to be moved left.
>
> **Full DP.** references are:
>
> [1] Density estimation: Duchi, John C., Michael I. Jordan, and Martin J. Wainwright. "Minimax optimal procedures for locally private estimation." Journal of the American Statistical Association 2018.
>
> [2] Classification: Berrett, T., C. Butucea. Classification under local differential privacy. arXiv:1912.04629, 2019.
>
> [3] Regression: Berrett, T. B., L. Györfi, H. Walk. Strongly universally consistent nonparametric regression and classification with privatised data. Electronic Journal of Statistics, 15:2430–2453, 2021.
>
> **Non-private.** There are many related references. Here we only show some standard literatures.
>
> [4] K. Chaudhuri et al. "Rates of convergence for nearest neighbor classification." NeurIPS 2014.
>
> [5] J. Audibert et al.. "Fast learning rates for plug-in classifiers." Annals of Statistics, 2007.
>
> [6] P. Zhao et al. "Minimax rate optimal adaptive nearest neighbor classification and regression." IEEE transactions on information theory, 2021.
>
> **In the “Minimax analysis for private data” paragraph, KNLRS11 is attributed with finding the relation between label DP and stochastic queries. This is not accurate, this work characterizes local DP learning by the statistical query model.**
>
> Thanks. It is a typo here. We actually mean "finding the relation between local DP and statistical queries." "label" should be "local", "stochastic" should be "statistical".
>
> **“We hope that to be as small as possible” -> “we seek to minimize this risk” or something similar**
>
> Thanks. "we seek to minimize the risk" is indeed better and we will change it during revision.
>
> **In Proposition 2, f(x) is used before it is defined.**
>
> Thanks. f(x) is defined in Assumption 1(c) now. We should place it within Proposition 2.
>
> "I didn’t find the proof outline for Theorem 1 or Theorem 3 to be informative at all. It would be good to add more specifics if possible."
>
> Thanks. This point is also raise by reviewer #26J5. We refer to the answer to question 1 and 4 there.
>
> Theorem 1 is relatively simple. It uses existing techniques for standard non-private minimax analysis, as well as standard analysis for local DP (see question 1 in the reply of #26J5). Compared with existing works, we need to bound the divergences between two joint distributions with both public and private parts.
>
> For Theorem 3, the first half and second half of the paragraph (line 214-219) are not connected. We will only expand the second half in the revised paper.
>
> Similar to classical minimax theory, we derive the lower bound by dividing the support into $G$ bins. For each bin, we construct a binary hypothesis. The lower bound of excess risk can then be converted to the lower bound of the error of hypothesis testing. Towards this goal, we develop a new tool (Lemma 1) to derive such a lower bound.
>
> **Do you have a sense of how the rates would change for approximate label DP?**
>
> Thanks for raising this good question. For approximate $(\epsilon, \delta)$-DP, we usually consider the case with very small $\delta$. Our intuition is that the rates will not be significantly different from pure DP (with small $\delta$). However, currently it seems that there are no effective approaches to derive the minimax lower bound with approximate local DP. Therefore, currently we can not make rigorous statements.

---

> > ### Comment · Reviewer_vnjU · 2024-08-12
> >
> > Thank you to the authors for addressing my concerns and committing to improving the presentation of the paper. My positive score was motivated by the technical contribution and the assumption that presentation issues would be addressed during the review process and so I will keep my score.

---

> > > ### Author Response · Authors · 2024-08-13
> > >
> > > Thank you very much for your response. We are glad that you maintain the score. Please let us know if you have any further questions or comments.

---

### Official Review · Reviewer_26J5 · 2024-07-29

**Soundness:** 4
**Presentation:** 2
**Contribution:** 3
**Rating:** 5
**Confidence:** 3

**Summary:**

The paper considers the problems of classification and regression under the constraint of local/central pure label DP. The authors derive upper and lower bounds on the excess risk (compared to the non-private Bayes classifier/regression) for these problems, under somewhat standard assumptions on the 'ground truth' randomized label function $\eta$. For regression, both the case where the labels are bounded and have bounded moments are considered. For the lower bounds, the authors develop extensions of techniques from minimax estimation to label DP. For upper bounds, authors propose some algorithms combining 'binning' different examples with a privacy mechanism chosen according to the problem setting. The upper/lower bounds are matching in each setting up to logarithmic factors. For local label DP, the authors show the minimax excess risk with $N$ samples matches the non-private bounds using $N \min\{\epsilon^2, 1\}$ samples. In other words, with $\epsilon = \Omega(1)$ the minimax risk asymptotically matches the non-private risk, and otherwise there is an inherent separation. For central label DP, the minimax bound is one that approaches the non-private bound as $N \rightarrow \infty$ for any fixed $\epsilon$, showing a qualitative difference. For local "full" DP, i.e. the features are also private, even for $\epsilon = \Omega(1)$ and large $N$ one cannot achieve the non-private rate.

**Strengths:**

* Derives optimal (up to log factors) upper and lower bounds for several different variants of classification/regression under label DP.
* To derive these bounds, introduces some new technical tools for minimax analysis of DP algorithms that might be useful in future work.
* Label DP is a variant of DP that is seeing attention in practice, and classification/regression are fundamental problems, so the results in the paper can have a practical impact easily.
* The authors do a good job making clear the comparison between the results in different settings. e.g. Table 1 is a very concise summary that allows one to draw all the essential comparisons between the different settings, and there are discussions like Remark 1 that give qualitative interpretations of the quantitative results, and also discuss other baselines to compare to.

**Weaknesses:**

The main issue is with the presentation. Specifically, the presentation does a great job explaining what the final results are and helping the reader contextualizing them, but at some points the techniques used to obtain the result are discussed at a very high level in the main body and why they work remains obscure even after reading the proof outlines in the main body multiple times.  There are some cases where the authors do a good job concisely describing a proof, e.g. Theorem 6's proof outline is very concise but it still gives a good idea what the proof looks like, even if they would have to check the appendix for details. But for others, like Theorems 1/2/3, the proof outline is not very informative. See Questions for more details.

I understand the authors are constrained by space requirements, but I think the allocation of space in the main body can be better thought out. For example, I think it might be better to try to give the reader a very good understanding of classification and/or bounded label regression (e.g., Lemma 1 from the Appendix could be brought to the main body without its proof, and the authors could explain how it is used), and omit all but the top-level points on bounded label moment regression, rather than giving a sparse understanding of all three.

**Questions:**

These are some examples of points I think are not clear from the main body which a reader interested in these problems might be able to understand better if the presentation were improved.

* What techniques from local DP / minimax theory does Theorem 1 use / how would it vary from e.g. the proof for the corresponding non-private lower bound? Saying you use techniques from certain areas/papers without specifying what they are is not particularly informative to a reader. (even giving a high-level summary of the technique would be ok)
* How do you decide the number of bins in your upper bound for Theorem 2? Presumably by optimizing some tradeoff b/t the number of examples per bin and having narrower bins where $\eta$ cannot vary too much within a bin, but I'm not confident about this from reading the main body.
* $\mathcal{X}$ is a vector space, what does "a bin of length $h$" mean for bins defined over a vector space? I believe it means something like each bin has radius at most $h$ in some appropriate norm, but not sure.
* In Theorem 3, the proof outline's first and second halves seem disconnected. The authors mention the model complexity needs to increase with $N$, how does the second half of the proof outline address this necessity?

**Limitations:**

Yes

---

> ### Author Rebuttal · Authors · 2024-08-01
>
> Thank you very much for your valuable suggestion on improvement of presentation.
>
> We agree that Lemma 1 in the appendix is important. In our revised paper, we will move this lemma to the main body of the paper. Moreover, we will also provide a better idea of the proof.
>
> The outline of proving theorem 1, 2 and 3 can be written in another way. In particular, in the revised paper, we will add some discussions (for the contents, see the answers of question 1 and 4 below).
>
> **Question 1: What techniques from local DP / minimax theory does Theorem 1 use / how would it vary from e.g. the proof for the corresponding non-private lower bound? Saying you use techniques from certain areas/papers without specifying what they are is not particularly informative to a reader. (even giving a high-level summary of the technique would be ok)**
>
> (1) **Techniques from non-private minimax theory.** In minimax lower bound, one takes minimum of all methods and maximum over all possible distributions. We can then narrowing the whole set of distribution to a small set parameterized by a binary vector $v\in \\{-1,1\\}^m$. Similar analysis can be found in the proof of Theorem 6 in [1], the proof of Theorem 3.5 in [2] and proof of Theorem 2 in [3].
>
> [1] K. Chaudhuri et al. "Rates of convergence for nearest neighbor classification." NeurIPS 2014.
>
> [2] J. Audibert et al.. "Fast learning rates for plug-in classifiers." Annals of Statistics, 2007.
>
> [3] P. Zhao et al. "Minimax rate optimal adaptive nearest neighbor classification and regression." IEEE transactions on information theory, 2021.
>
> These discussions are also used to reply to reviewer kpNG.
>
> (2) **Techniques from local DP.** We use Theorem 1 in [4], in (c) in eq.(42).
>
> [4] Minimax optimal procedures for locally private estimation. Journal of the American Statistical Association. 2018
>
> (3) **New techniques that are not from either local DP or classical minimax theory.**  The label local DP problem lies in between the full DP and non-private case, so we can not directly bound the divergence based on existing techniques. In this paper, we design a new bound for the divergence between two joint distributions that have both private and non-private parts. See eq.(46) and eq.(47) in the paper.
>
> Theorem 1 is relatively easier than later theorems. We will also provide more discussion about other theorems in the paper.
>
> **Question 2: How do you decide the number of bins in your upper bound for Theorem 2? Presumably by optimizing some tradeoff b/t the number of examples per bin and having narrower bins where $\eta$ cannot vary too much within a bin, but I'm not confident about this from reading the main body.**
>
> Yes, the number of bins is selected to achieve a bias variance tradeoff. Too large bins will induce strong bias. Too narrow bins will make the random fluctuation of $\eta$ be high.
>
> For the optimal number of bins, in line 189, we have determined $h$. Note that the volume of each bin is $h^d$, thus $G=V/h^d$, with $V$ being the total volume of support $\mathcal{X}$. Therefore $G\sim (N(\epsilon^2\wedge 1)/ln K)^{d/(2\beta+d)}$.
>
> **Question 3. $\mathcal{X}$ is a vector space, what does "a bin of length $h$" mean for bins defined over a vector space? I believe it means something like each bin has radius at most $h$ in some appropriate norm, but not sure.**
>
> Here $\mathcal{X}$ is a compact set. Note that assumption 1(c) requires that $f(x)\geq c$, which means that $f$ is bounded away from zero. The pdf is normalized to 1, thus the support must be bounded. $\mathcal{X}$ is segmented into many cells, and the length of each cell is $h$. We will clarify it in our revised paper.
>
> **Question 4. In Theorem 3, the proof outline's first and second halves seem disconnected. The authors mention the model complexity needs to increase with $N$, how does the second half of the proof outline address this necessity?**
>
> We agree that the first and second halves are disconnected. The first half is used to argue that the proof can not simply use existing methods (i.e. packing method), since packing method is only suitable for problems with fixed complexity. Therefore, this problem requires new technical novelties.
>
> We think that it would be better to put the first half below, in order to avoid confusions.
>
> In general, we thank again for these valuable suggestions on improving the readability of this paper. We will revise this paper accordingly.

---

> > ### Comment · Reviewer_26J5 · 2024-08-11
> >
> > Thanks for the response. I am hopeful that the authors can incorporate the ideas presented in the rebuttal to improve the presentation. At this point, I agree with the sentiment of Reviewer kpNG - I do not oppose accepting the paper, though a round of reviews to help validate that the presentation has improved may be good. So I am keeping my score.

---

> > > ### Author Response · Authors · 2024-08-12
> > >
> > > Thanks for your response. As the final version allows one additional page, there will be enough space for us to provide more detailed and informative outlines for each theorem. Please kindly let us know if you have further questions or comments. We will add all the new results and discussions to the final paper. Thank you very much!

---

### Official Review · Reviewer_kpNG · 2024-07-29

**Soundness:** 2
**Presentation:** 2
**Contribution:** 2
**Rating:** 4
**Confidence:** 2

**Summary:**

This paper investigates the minimax risks of classification and regression (with both bounded and heavy-tailed noise) under label differential privacy (DP) in both central and local models.

**Strengths:**

The paper provides a comprehensive analysis by considering both upper and lower bounds for the minimax risks.
It explores both central and local DP models and different settings, covering a broad spectrum of scenarios.

**Weaknesses:**

The writing quality needs improvement to meet publication standards. Several sections are challenging to understand. Specific issues include:
(1) Around line 178, the output of the mechanism for classification is unclear. Why is it not a one-hot vector, or at least why is the L1 norm not equal to 1?
(2) Some notations are overused. For example, "c" refers to the lower bound of the density function in Assumption 1 and also denotes the classifier in line 186 and subsequent proofs.
(3) The description of the algorithm before Theorem 2 is vague and lacks clarity.
(4) The proofs in the appendix are hard to follow without explanations or discussions. For instance, how is $\phi$ defined in Equation (35), and what purpose does it serve? Why does the construction satisfy the assumptions? There seem to be some typos or missing elements in Equations (39) and (40).

**Questions:**

The paper discusses non-private baselines. Are there citations provided for these baselines?

**Limitations:**

See weaknesses and questions.

---

> ### Author Rebuttal · Authors · 2024-08-01
>
> Thanks for your review and valuable suggestions. We will update our paper in our revised version accordingly.
>
> **1. Around line 178, the output of the mechanism for classification is unclear. Why is it not a one-hot vector, or at least why is the L1 norm not equal to 1?**
>
> As long as the privacy requirement is satisfied, it is not required to conduct one hot encoding. One hot encoding will induce unnecessary loss of information. If we use standard randomized response:
>
> $P(M(x,y)(j)=1)=e^\epsilon/(e^\epsilon + K - 1)$ if $y=j$,
>
> and
>
> $P(M(x,y)(j)=1)=1/(e^\epsilon + K - 1)$ if $y=j$,
>
> then with the increase of $K$, the probability of remaining the original label will decrease significantly. As a result, the classification risk will also increase.
>
> Moreover, this privatization mechanism is similar to RAPPOR [1].
>
> Also, the $\ell_1$ norm is not necessarily 1 since we use sigmoid instead of softmax activation in the last layer.
>
> **2. Some notations are overused. For example, "c" refers to the lower bound of the density function in Assumption 1 and also denotes the classifier in line 186 and subsequent proofs.**
>
> Thanks! There is indeed an overuse of notation $c$. We will use different notations in our revised paper.
>
> **3. The description of the algorithm before Theorem 2 is vague and lacks clarity.**
>
> Thanks for pointing out these. We thought that this part is easier than later theorems, due to space limitation, we only describe the algorithm in a concise way.
>
> Here we explain the idea of eq.(11) again. The label can be one hot encoded to a vector first. However, this is not differentially private. To meet the privacy requirement, we randomly flip each element in the vector with probability $1/(e^{\epsilon / 2}+1)$. The vector after such random flipping satisfies $\epsilon$-local label DP.
>
> Divide the support into $G$ bins. For each bin, we count the number of samples (denoted as $S_{lj}$) with $Z_i(j)=1$ and $X_i\in B_l$. Such number reflects the conditional probability of the label being $j$, given the feature $x\in B_l$. Therefore, the classification rule is to take argmax of $S_{lj}$ .
>
> **4. The proofs in the appendix are hard to follow without explanations or discussions. For instance, how is $\phi$ defined in Equation (35), and what purpose does it serve? Why does the construction satisfy the assumptions? There seem to be some typos or missing elements in Equations (39) and (40).**
>
> Thanks the reviewer for reading the appendix carefully. Although we are sure about the correctness of the proof, we also agree that are indeed some points that require more explanations.
>
> **(1) About $\phi$.**   As stated in line 489 and eq.(35), $\phi$ can be any function satisfying $\beta$-Holder assumption, i.e. $|\phi(x)-\phi(x')|\leq L||x-x'||^\beta$ which is supported on $[-1/2,1/2]^d$, and $0\leq \phi\leq 1$. Here it is not necessary to specify the form of $\phi$ exactly, as infinite number of possible functions $\phi$ are enough for our proof.
>
> Our construction satisfies Assumption 1. For (a), i.e. the Holder assumption with parameter $\beta$, from the construction eq.(36), $|\eta_v(x)-\eta_v(x')|\leq |\phi(\frac{x-c_k}{h})-\phi(\frac{x'-c_k}{h})|h^\beta\leq L(||x-x'||/h)^\beta h^\beta\leq L||x-x'||^\beta$, satisfying Assumption 1(a).
>
> For (b), from (36), $|\eta_v(x)|\leq h^\beta$. Therefore Assumption (b) holds for $t>h^\beta$. From eq.(37), $m\lesssim h^{\gamma \beta-d}$, we have $mh^d\lesssim h^{\gamma \beta}=(h^\beta)^\gamma$, indicating that Assumption (b) holds for $t=h^\beta$. The case with $t<h^\beta$ can be proved by the continuity of function $\phi$.
>
> Our construction is common in minimax analysis in nonparametric statistics. We refer to the proof of Theorem 6 in [2], the proof of Theorem 3.5 in [3] and proof of Theorem 2 in [4] for other examples that use similar ideas.
>
> (2) About eq.(39) and eq.(40). Thanks for pointing it out. In (39), $P(sign(\hat{\eta}(x)))$ should change to $P(sign(\hat{\eta}(x))\neq  \eta_v(x))$, which means the probability of making wrong classification at $x$. The typo in (40) is similar.
>
> We thank the authors for the suggestion. We will add more discussions in the revised version.
>
> **Questions. The paper discusses non-private baselines. Are there citations provided for these baselines?**
>
> The related works on non-private classification and regression are summarized in line 73-78 in the paper.
>
> (1) Classification. We refer to [2], theorem 4(b), which gives $O(n^{-\alpha(\beta+1)/(2\alpha+1)})$. Here the notations are different: $\alpha$ in [2] corresponds to $\beta/d$ in our paper, and $\beta$ in [2] corresponds to $\gamma$ in our paper. After such adjustment, it becomes $n^{-\beta(\gamma+1)/(2\beta+d)}$ in our paper.
>
> (2) Regression. Without privacy constraints, there are no significant differences between bounded and unbounded label noise. We refer to Theorem 5 in [4]. [4] considers heavy-tailed distributions, while current works focus on the bounded case, thus $\beta$ in Theorem 5 in [4] can be regarded as infinite. $p$ in [4] corresponds to $\beta$ in our paper.
>
> Nonparametric classification and regression have been widely studied and the related works are far beyond those listed above. We refer to a book [5] for a complete overview of nonparametric classification and regression.
>
>  In our revised version, we will cite related works directly in the table.
>
>
>
> # References
>
> [1] P. Kairouz et al. "Discrete distribution estimation under local privacy." ICML 2016.
>
> [2] K. Chaudhuri et al.  "Rates of convergence for nearest neighbor classification." NeurIPS 2014.
>
> [3] J. Audibert et al.. "Fast learning rates for plug-in classifiers." Annals of Statistics, 2007.
>
> [4] P. Zhao et al. "Minimax rate optimal adaptive nearest neighbor classification and regression." IEEE transactions on information theory, 2021.
>
> [5] Tsybakov. "Introduction to nonparametric estimation". 2009

---

> > ### Comment · Reviewer_kpNG · 2024-08-07
> >
> > Thank you to the authors for the detailed response. After reviewing other comments, it appears that many reviewers share concerns about the presentation, which makes the paper difficult to read, especially for those unfamiliar with the existing literature. There are discussions in the paper whose correctness I am unable to verify or appreciate due to my unfamiliarity with the existing literature and the poor presentation. If the consensus is that these flaws do not overshadow the merits of the content, I would not oppose its acceptance. However, I also support the idea of rejecting this version to allow for further refinement and improvement in a future submission.

---

> > > ### Author Response · Authors · 2024-08-08
> > > **Thanks for your reply**
> > >
> > > Thanks for your prompt response. Our feeling is that in general, reviewers raise good comments, indicating that they understand our paper well. The issues in presentation do not affect the readability significantly. We will definitely fix these issues in our revision.

---

> > > > ### Comment · Reviewer_kpNG · 2024-08-12
> > > >
> > > > Thank you for your response. The other reviewers indeed provided valuable feedback, indicating a better understanding of the paper. I feel that my difficulty in understanding some aspects may partially stem from my unfamiliarity with the related work. In response, I’ve decided to raise my score and moderately lower my confidence level. However, I believe that the presentation issues should be addressed in the revision to improve readability.

---

> > > > > ### Author Response · Authors · 2024-08-12
> > > > >
> > > > > Thank you very much for raising the score! Although other reviewers also raise some detailed issues, from their comments, we think that they have understood most of the paper well. We will definitely revise the details of presentations according to these comments. Moreover, in the final version, an additional page is allowed, thus we can add more discussions about the intuitions.
> > > > >
> > > > > Please let us know if you have any further concerns.

---

### Author Rebuttal · Authors · 2024-08-05

Thanks all the reviewers for your careful reading and valuable comments. We are encouraged to know that reviewers are positive about the novelty and value of our works. We have also received some detailed comments about definitions, notations and presentations that can be further improved. Some of them are raised by more than one reviewers. Thank you very much for these comments! We will definitely revise the paper based on these comments.

---

### Decision · Program_Chairs · 2024-09-25

**Decision:**

Reject

**Comment:**

This paper studies the classification and regression under label differential privacy (label DP) under the minimax framework. Here, we are given labeled samples $(X_1, Y_1), \\dots, (X_n, Y_n)$ drawn from some unknown distribution $D$ and the goal is to build a classification or regression model that minimizes the expected loss w.r.t. $D$. The samples $X_1, \\dots, X_n$ here are assumed to be public while the labels $Y_1, \\dots, Y_n$ are assumed to be private, i.e. the neighboring notion for label DP is changing a single label. The main contribution of the paper is to, under standard assumptions (e.g. that the underlying label probability is Holder continuous), give tight minimax bounds for classification and regression with label DP both under local DP and central DP. For central DP, there is almost no cost for label DP (compared to no privacy constraint) as long as $\\epsilon$ is not very small. Meanwhile, there is a cost of local label DP as long as $\\epsilon \ll 1$.

**Strengths:**
- This paper gives tight minimax bounds for classification and regression with label DP under standard assumptions.

**Weaknesses:**
- Although the bounds are nice, it is very hard to tell from the main body what the novel ideas are. More specifically, the "proof outlines" for the lower bounds essentially contain no information. Meanwhile, the upper bounds simply use binning together with standard DP algorithms, i.e. RAPPOR in equation (11), Exponential Mechanism in equation (17) and Laplace mechanism (for regressions).

- Although this paper seems to be the first to study classification / regression under Assumption 1, there were numerous previous works that derive theoretical (lower and upper) bounds for classification / regression under label DP before. For example, the paper [14] provides upper & lower bounds for label-DP convex optimization; a much earlier paper [A] also gave PAC-style bounds for learning. A more comprehensive discussion is required to distinguish the current work from previous works.

- The paper only deals with pure-DP and does not consider approximate-DP.

**Recommendation** The paper would benefit from a significant revision to highlight the novelty, and to provide comparisons to previous work. Due to this, we recommend rejection for the current version of the paper.

**References:**

[A] Kamalika Chaudhuri, Daniel J. Hsu: Sample Complexity Bounds for Differentially Private Learning. COLT 2011: 155-186